# CAN LLMS EVALUATE COMPLEX ATTRIBUTION IN QA? AUTOMATIC BENCHMARKING USING KNOWLEDGE GRAPHS

## ABSTRACT

The attribution of question answering (QA), which is to get evidences for supporting the generated answer, has attracted wide research attention. The current methods for automatically evaluating the attribution, typically relying on Large Language Models (LLMs), are still inadequate, particularly in recognizing subtle differences between attributions, and in measuring complex attribution reasoning. Existing benchmarks, which are primarily based on manual annotations, suffer from limited evaluation settings with incomplete and coarse attribution categories and reasoning scenarios, hindering the evaluation and advancement of attribution evaluators. To address this gap, we introduce Complex Attributed Question Answering (**CAQA**), a large-scale benchmark automatically generated using Knowledge Graphs (KGs), containing more comprehensive attribution categories and complex attribution reasoning scenarios. Our experiments with two specifically developed evaluators and nine LLM evaluators reveal that they struggle in identifying negative attribution categories and handling complex attribution reasoning in both zero-shot and few-shot settings, but mostly perform relatively well in the fine-tuning setting. Moreover, all evaluators perform inadequately in fine-grained attribution identification scenarios. The experiments also demonstrate that **CAQA** is consistent with human annotations, and is promising for selecting and developing more effective attribution evaluators in QA. The entire project is publicly accessible at `https://github.com/aannonymouuss/CAQA-Benchmark`.

## 1 INTRODUCTION

Generative AI (Brown et al., 2020; OpenAI, 2023; Touvron et al., 2023a) is increasingly adept together with other techniques like search engines to produce textual statements as answers to natural language questions. However, their tendency to generate confident yet inaccurate or "hallucinated" contents (Ji et al., 2023) poses significant risks in high-stakes domains such as medicine (Lee et al., 2023) and law (Volokh, 2023). In response to this challenge, question answering (QA) with attribution has been proposed, where not only answers but also citations (or evidence snippets) for supporting the answers are output (Menick et al., 2022; Rashkin et al., 2023; Bohnet et al., 2022; Li et al., 2023a). Such attributed models are essential for enhancing user trust and reliability of QA systems.

Despite their potential, state-of-the-art implementations of attributed QA, exemplified by generative Large Language Models (LLMs) with search engines like Bing Chat, perplexity.ai and YouChat[1], still often produce attribution errors (Liu et al., 2023). Therefore, it is crucial to explore effective automatic attribution evaluation methods, which can not only continuously measure the performance of attributed QA systems, but also provide feedback to improve their attributions (Yue et al., 2023; Gao et al., 2023a; Bohnet et al., 2022), alleviating the issues of factuality, faithfulness and hallucination (Amouyal et al., 2022; Asai et al., 2023). However, existing attributed QA benchmarks (as shown in Table 1) are inadequate in evaluating and advancing attribution evaluation methods due to their limited size and constrained evaluation settings. First, the attribution categories in these benchmarks lack comprehensiveness. Particularly, for the category `partially supportive`, no benchmark offers a fine-grained assessment, i.e. how many sub-facts in the answer can be supported by the

---

[1]bing.com/new, perplexity.ai, https://you.com/

evidence. Second, these benchmarks ignore the reasoning complexity in judging attributions that require reasoning with multiple pieces of evidence under various logical combinations. Such complex attributions are quite common in Bing Chat and retrieve-and-read systems (Malaviya et al., 2023).

In this work, we introduce a comprehensive set of attribution categories for representing correct and different kinds of incorrect attribution cases: `supportive`, `partially supportive`, `contradictory` and `irrelevant` (see Table 2 for examples). We also define different levels of attribution complexity based on the reasoning logic required to infer the answer by the evidence: `single`, `union`, `intersection`, and `concatenation` (see Table 3 for examples). Based on these, we construct the Complex Attributed Question Answering (CAQA) benchmark to compare attribution evaluation methods and develop better ones. Compared with existing benchmarks (see Table 1), CAQA features a larger scale, more comprehensive attribution categories, and varying levels of attribution complexity. Significantly, it is the only benchmark to provides a fine-grained evaluation for the `partially supportive` scenario. To construct this benchmark, we introduce an automatic generation method based on a Knowledge Graph (KG) (Hogan et al., 2021; Bollacker et al., 2008), which is composed of relational facts in the form of triples, and two KGQA datasets, containing question-answer pairs and corresponding KG queries. Our method extends these queries using various rules that introduce additional logical operators to increase reasoning complexity. These extended queries are then employed to extract KG sub-graphs, which are edited using different strategies to create diverse attribution categories. Finally, the edited sub-graphs are transformed into natural language citations using ChatGPT prompting. This approach is flexible, allowing the generation of attributed QA benchmarks with varied features, and adaptable to different KGs and KGQA datasets.

Table 1: Comparison of CAQA with other benchmarks. Category denotes the attribution categories in each benchmark, including Supptive (S), Non-supportive (N), Partially Supportive (P), Contradictory (C), Irrelevant (I) and Extrapolatory (E), with E and I treated as equivalent. Comp. denotes whether the benchmark contains a reasoning complexity classification for attribution, and Auto. indicates the benchmark is automatically constructed without manual annotation.

| Benchmarks | #Sample | Category | Comp. | Auto. |
|---|---|---|---|---|
| Bohnet et al. (Bohnet et al., 2022) | 23,000 | S/N | ✗ | ✗ |
| HAGRID (Kamalloo et al., 2023) | 2,638 | S/N | ✗ | ✗ |
| ExpertQA (Malaviya et al., 2023) | 2,177 | S/N | ✗ | ✗ |
| AttributionBench (Li et al., 2024) | 17,816 | S/N | ✗ | ✗ |
| Liu et al. (Liu et al., 2023) | 11,037 | S/P/N | ✗ | ✗ |
| ALCE (Gao et al., 2023b) | 800 | S/P/N | ✗ | ✗ |
| AttrEval-Gen (Yue et al., 2023) | 242 | S/C/E | ✗ | ✗ |
| AttrEval-Sim (Yue et al., 2023) | 64.2K | S/C/E | ✗ | ✓ |
| CAQA (Ours) | 161.1K | S/P/C/I | ✓ | ✓ |

We evaluate two particularly developed evaluators (fine-tuned on specific data) and nine LLM evaluators under zero-shot, few-shot and fine-tuning settings. Here are some of the important observations. (1) All evaluators struggled to identify the nuanced negative attribution categories in both zero-shot and few-shot settings. For example, the highest F1 score of recognising `partially supportive` is only 45.6% (reps. 53.9%) under the zero-shot (resp. few-shot) setting. With fine-tuning, the F1 scores of all the categories exceed 90% for most LLM evaluators. Moreover, all evaluators perform poorly in the fine-grained evaluation of "partially supportive", while those who could only identify coarse attribution categories perform better. (2) Evaluators perform worse on more complex attribution categories such as `concatenation` and `intersection`, which require more advanced logical reasoning. (3) When tested on an out-of-distribution dataset, LLM evaluators fine-tuned by our CAQA dataset achieve better performance than the particularly developed evaluators. This result highlights the potential of the CAQA for training more effective evaluators for attributed QA.

## 2 RELATED WORK

**Attributed Question Answering.** Generative LLMs now lead the performance in QA, but often produce hallucinations (Ji et al., 2023; Xiao & Wang, 2021; Wang & Sennrich, 2020; Shuster et al., 2021). To alleviate this issue, some studies (Menick et al., 2022; Nakano et al., 2021; Gao et al., 2023b) train attributed models to answer questions while supporting attribution with citations and references. Some other studies augment LLMs with external tools (Mialon et al., 2023; Shen et al., 2023; Schick et al., 2023) such as retrievers (Han et al., 2023; Shi et al., 2023; Asai et al., 2023; Izacard et al., 2022) and search engines (Nakano et al., 2021; Komeili et al., 2021), or incorporate external references for attribution. However, the quality of such attributions remains questionable, and their automatic evaluation is still an open research question.

Table 2: Examples of the four attribution categories. Green, yellow, and red text indicate the content in the answer that is supported, not supported, or contradicted by the content in the citation, respectively.

| Attribution Category | Examples |
|---|---|
| **Supportive** | **Question:** Who plays Fruma Sarah in Fiddler on the Roof?
**Answer:** Fruma Sarah is a character in the musical "Fiddler on the Roof'", and Ruth Madoc played the role [1].
**Citations:** [1] ... In 1971 Ruth Madoc played Fruma Sarah in the film version of the musical "Fiddler on the Roof", and in 1972 she appeared as ... |
| **Partially Supportive** | **Question:** Who plays Patrick in 10 Things I Hate About You?
**Answer:** Patrick is played by actor Heath Ledger in the film 10 Things I Hate About You [1].
**Citations:** [1] 10 Things I Hate About You is a 1999 American teen romantic comedy-drama film directed by Gil Junger and starring Heath Ledger, Julia Stiles, Joseph Gordon-Levitt, and Larisa Oleynik. The screenplay, written by ... |
| **Contradictory** | **Question:** Who directed a George Pal's production?
**Answer:** George Pal directed a production called Puppetoons [1].
**Citations:** [1] ... The Puppetoon Movie is a 1987 animated film written, produced, and directed by Arnold Leibovit ... |
| **Irrelevant** | **Question:** Who played the weasley brothers in Harry Potter?
**Answer:** James and Oliver Phelps, identical twin actors, played the roles of Fred and George Weasley in the Harry Potter film series [1].
**Citations:** [1] Chris Rankin plays of "Bugsy Malone", "The Lion, The Witch and The Wardrobe" and Harry Potter series ... he plays a brother of Harry Potter's best friend, ... |

**Attribution Evaluation.** Current methods for evaluating attribution predominantly depend on human annotation (Nakano et al., 2021; Bohnet et al., 2022; Liu et al., 2023; Rashkin et al., 2023; Muller et al., 2023), which is costly and very inefficient. Recent studies propose automatic attribution evaluators based on LLMs, such as AUTOIS (Gao et al., 2023a; Bohnet et al., 2022) and ATTRSCORE (Yue et al., 2023). However, existing attributed QA benchmarks are inadequate for evaluating and advancing attribution evaluators due to their limited size and restricted evaluation settings, including incomplete attribution categories and omission of reasoning complexity in judging attributions. Most benchmarks classify attribution into only two categories: the cited evidence *supports* or *does not support* the answer (Gao et al., 2023b; Li et al., 2023b; 2024; Malaviya et al., 2023; Bohnet et al., 2022). Some benchmarks (Gao et al., 2023b; Liu et al., 2023; Zhang et al., 2024) add a third category, *partially supportive*, but their sizes are small and reliance on manual annotation. Yue et al. (2023) presents a method for automatically generating attribution annotations to construct large-scale samples with categories of *supportive*, *contradictory*, and *extrapolatory* (equivalent to *irrelevant*). However, their method cannot generate the *partially supportive* category, as it relies solely on answer word replacement to construct other categories. Our work addresses these limitations by proposing a novel method based on knowledge graphs (KGs) and knowledge graph question answering (KGQA) datasets to automatically create a large-scale attribution QA benchmark with comprehensive attribution categories. Notably, our benchmark is the first to offer fine-grained evaluation for partially supportive scenarios and considers varying levels of logical reasoning complexity in attribution.

## 3 DEFINITIONS IN QUESTION ANSWERING ATTRIBUTION

### 3.1 TASK FORMULATION

This work studies the task of evaluating attributed QA. It is to verify whether an evidence, which has one or multiple citations (references) with facts stated, can sufficiently support a generated answer statement towards a natural language question. Formally, given a question $q$, an answer statement $a$ and an evidence $e$, the objective of attribution evaluation is to map them to an attribution category $t$ (a.k.a. class label). Note that $q$, $a$ and $e$ are all in natural language. This mapping can be represented by the function $\mathcal{F}: \mathcal{Q} \times \mathcal{A} \times \mathcal{E} \mapsto \mathcal{T}$, where $\mathcal{Q}$, $\mathcal{A}$ and $\mathcal{E}$ denote the sets of questions, answers and evidences, respectively, and $\mathcal{T}$ denotes the set of potential categories, such as {*supportive*, *partially supportive*, *contradictory*, *irrelevant*} which mean "the evidence $e$ is supportive, partially supportive, contradictory or irrelevant to the fact that $a$ is the answer of the question $q$."

Table 3: Examples of the four complexity types. Reasoning Graphs show the reasoning relationships between citations-answers. Green represents content associated with the answer, gray indicates excluded content, and orange indicates the common term connecting the citations.

| Complexity | Examples | Reasoning Graphs |
|---|---|---|
| **Single** | **Question:** Which radio program episode appears in All Things Considered? **Answer:** The radio program episode in which All Things Considered appears is Remorse: The 14 Stories of Eric Morse [1]. **Citations:** [1] *Remorse: The 14 Stories of Eric Morse is an episode of the radio program All Things Considered*.... | $C_{[1]}$ —Support→ Ans |
| **Union** | **Question:** Which university did Rick Scott attend? **Answer:** Rick Scott attended the University of Missouri–Kansas City and Southern Methodist University [1][2]. **Citations:** [1] *Rick Scott graduated from the University of Missouri–Kansas City* ... [2] *Rick Scott earned a juris doctor degree by working his way through Southern Methodist University*, ... | $C_{[1]}$ $C_{[2]}$ —Support→ Ans |
| **Intersection** | **Question:** The computer designer for Macintosh 128k and NeXT computer was whom? **Answer:** The computer designer for Macintosh 128k and NeXT computer was Steve Jobs [1][2]. **Citations:** [1] *The computer designer for Macintosh 128k was* Jerry Manock, *who worked with Steve Jobs to develop the vertical body* ... [2] ...*Several former Apple employees followed Jobs to NeXT*, *including Joanna Hoffman, Bud Tribble, George Crow, Rich Page*... | $C_{[1]}$ $C_{[2]}$ —Support→ Ans |
| **Concatenation** | **Question:** What are the official languages in the politician Mohammad Najibullah's country? **Answer:** Pashto and Dari are the official languages in the politician Mohammad Najibullah's country. [1][2]. **Citations:** [1] *Mohammad Najibullah was the president of Afghanistan from 1986 to 1992* ... [2] *Afghanistan* s *a multilingual country, where Pashto and Dari (a dialect of Persian) are the official languages with* ... | $C_{[1]}$ $C_{[2]}$ —Support→ Ans |

## 3.2 FINE-GRAINED ATTRIBUTION CATEGORIZATION

We analyse the results of practical attributed QA systems (Gao et al., 2023b) and find that apart from correct attributions *supportive*, there are three main causes of incorrect attributions: *partially supportive*, *contradictory* and *irrelevant*. More details are shown in Appendix F. The four attribution categories are defined below:

- **Supportive (Sup.)**: The evidence includes facts that can fully support the answer statement.
- **Partially Supportive (Par.)**: The evidence lacks a part of the facts that are required to infer the answer statement.
- **Contradictory (Con.)**: The evidence includes facts that can infer a different answer statement.
- **Irrelevant (Irr.)**: The evidence has no facts that can be used to infer the answer statement.

Table 2 provides examples of the four attribution categories. In the **supportive** scenario, the answer is backed by citation [1], which confirms that "*Ruth Madoc plays Fruma Sarah in Fiddler on the Roof.*" In the **partially supportive** scenario, the answer cites [1] but does not fully align with the complete context provided, mentioning only "*the actor Heath Ledger stars in the film 10 Things I Hate About You*" and missing the information "*Heath Ledger plays the character Patrick*". **Note that the partially supportive scenario in our benchmark supports fine-grained evaluation**, assessing how many sub-facts in the answer can be supported by the citation. For example, the answer contains the sub-facts [*Patrick, played_by, Heath Ledger*] and [*Heath Ledger, star_in, 10 Things I Hate About You (film)*], but only the latter sub-fact is supported by the citation. In the **contradictory** scenario, the citation [1] states "*The Puppetoon Movie is directed by Arnold Leibovit*," which contradicts the generated answer. The **irrelevant** scenario involves citing [1], which discusses an unrelated actor, Chris Rankin, and his career offers no relevant facts to verify the answer.

## 3.3 ATTRIBUTION COMPLEXITY

Previous research has not explored different levels of complexity in inferring the answer. Malaviya et al. (2023) has shown that AutoIS (Bohnet et al., 2022), the most commonly used automatic attribution evaluation method, often mistakes in scenarios that require multiple citations to validate

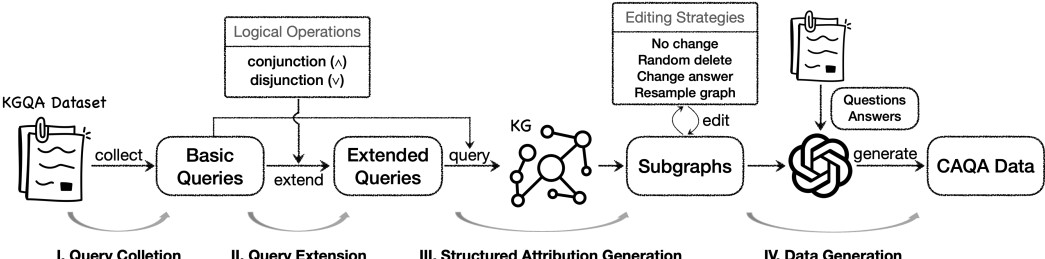

Figure 1: The entire process of constructing the CAQA benchmark.

the answer. To advance automatic evaluation methods, our benchmark incorporates reasoning complexity by categorizing attribution into four levels of complexity, based on the form of supporting facts in the citations (see Table 3 for examples):

- **Single (S.)**: The answer is supported by one fact from one single citation in the evidence.

- **Union (U.)**: The answer is supported by independent facts from multiple citations in.

- **Intersection (I.)**: The answer is supported by facts with some common entities from multiple citations.

- **Concatenation (C.)**: The answer is supported by chains of facts from multiple citations.

## 4 BENCHMARK CONSTRUCTION USING KNOWLEDGE GRAPH

In this section, we introduce our methodology that leverages KGs and KGQA datasets to construct attributed QA benchmarks. Figure 1 provides an overview of the benchmark construction process, which is comprised of four key steps:(1) Query Collection: Given a KGQA dataset, we collect data corresponding to three basic KG logical queries; (2) Query Extension: Two logical operators are applied to increase the complexity of the basic queries; (3) Structured Attribution Generation: The extended queries are grounded in the KG to obtain relevant subgraphs, which are then probabilistically edited using four strategies to generate new subgraphs with four attribution labels; (4) Data Generation: We produce attributed QA data, where each instance consists of an extended question, rephrased answer entities, citations derived from subgraphs, as well as attribution and complexity labels.

### 4.1 QUERY COLLECTION

We construct the attributed QA benchmark upon an existing KGQA dataset and its associated KG. This is primarily motivated by two observations: (1) KGQA is a well-established task with a wealth of open resources, as evidenced by 25 KGQA datasets for 5 KGs reported in (Jiang & Usbeck, 2022); (2) existing KGQA datasets contain high-quality question-answer pairs and corresponding KG logical queries, often expressed in SPARQL, which are capable of deriving the correct answers and can be leveraged to generate evidence.

The KG is composed of relational facts in the form of triple, i.e., $(h, r, t)$, where $h$ and $t$ denote a head entity (subject) and a tail entity (object), respectively, and $r$ denotes the relation between them. The KGQA dataset $D = \{S_1, S_2, ..., S_N\}$ consists of samples in the form of $S_i = (q_i, a_i, l_i)$, where $q_i$ denotes a natural language question, $a_i$ denotes its answer entity, and $l_i$ denotes the corresponding KG logical query of $q_i$. Our data collection focuses on samples where the KG logical query falls into one of three types: **single-triple**, **path-like**, or **tree-like queries**. As shown in the first three columns in Table 4, a single triple query denoted as $(e_0, r_0, ?a)$ indicates that the answer entity $?a$ can be obtained via the subject $e_0$ and the KG relation $r_0$. A path-like query denoted as $[e_0, r_0, ?v_1, \ldots, ?v_{n-1}, r_{n-1}, ?a]$ represents that the answer $?a$ is reachable through an $n$-hop path starting from subject $e_0$, traversing $n$ relations and $n-1$ intermediate entities. Notably, a path-like query reduces to a single-triple query when $n = 1$. Finally, a tree-like query, formulated as $\wedge_{i=0}^{n-1}(e_i, r_i, ?a)$, includes $n$ distinct triples, each originating from different subjects and converging on the same answer object $?a$.

Table 4: The rules for each type of original query $l$ to the extended query $l'$, utilizing two query operations: intersection ($\wedge$) and union ($\vee$). All queries are classified according to their structure as single-triple (S.) queries, path-like (P.) queries, tree-like (T.) queries and union-tree-like (U.) queries. The 'Examples' column presents corresponding graph representations for the case where $n = 2$, $m = 2$, and $k = 0$. In these graphs, grey nodes represent variables for answer entities, white nodes represent entities or variables for intermediate entities.

| Original Query $l$ | | | Extended Query $l'$ | | |
|---|---|---|---|---|---|
| **Definitions** | **Structures** | **Examples** | **Definitions** | **Structures** | **Examples** |
| $(e_0, r_0, ?a)$ | S. | | $(e_0, r_0, ?a)$ $\vee(e_1, r_0, ?a) \vee \ldots \vee (e_m, r_0, ?a)$ | U. | |
| $[e_0, r_0, ?v_1, \ldots, ?v_{n-1}, r_{n-1}, ?a]$ | P. | | $[e_0, r_0, ?v_1, \ldots, ?v_{n-1}, r_{n-1}, ?a]$ $\wedge(e_1, r_n, e_0)$ | P. | |
| | | | $[e_0, r_0, ?v_1, \ldots, ?v_{n-1}, r_{n-1}, ?a]$ $\wedge(e_1, r_n, ?a)$ | T. | |
| $\wedge_{i=0}^{n-1}(e_i, r_i, ?a)$ | T. | | $\wedge_{i=0}^{n-1}(e_i, r_i, ?a),\ i \neq k$ $\wedge(e_n, r_n, e_k) \wedge (e_k, r_k, ?a)$ | T. | |
| | | | $\wedge_{i=0}^{n-1}(e_i, r_i, ?a) \wedge (e_n, r_n, ?a)$ | T. | |

## 4.2 QUERY EXTENSION

For each KGQA example $S_i = (q_i, a_i, l_i)$, we extend one basic logical query $l_i$ to $l_i'$ using a set of predefined query extension rules. These rules are designed based on the logical operations *intersection* (a.k.a conjunction, $\wedge$) and *union* (a.k.a disjunction, $\vee$) (Ren et al., 2023)[2].

Table 4 outlines the extension rules. For a single-triple query $l$, the *union* operation is used. Initially, we retrieve entities from the KG that share the same name as $e_0$ in $l$, producing a set of $m$ entities $\{e_1, \ldots, e_m\}$, where $m$ may be zero. Subsequently, we generate logical queries $(e_1, r_0, ?a)$, ..., $(e_m, r_0, ?a)$ by combining the retrieved entities and the relation $r_0$ from $l$. These new queries are then merged with $l$ using the *union* operation, resulting in a union-tree-like query structure. This structure implies that the final answer is derived as the union of the answers obtained from each subquery. For a path-like query or a tree-like query, we apply the *intersection* operation in two distinct ways. In the first way, we identify a unique subject entity $e_0$ for path-like queries or randomly select a subject entity $e_k$ for tree-like queries. We then retrieve corresponding triples $(e_1, r_n, e_0)$ or $(e_k, r_n, e_n)$ from the KG, where $r_n$ represents a relation not present in $l$. These new triples are appended to the respective queries, ensuring that $e_0$ and $e_k$ are connected nodes. This process maintains the overall structure of the path-like or tree-like query. In the second way, we append a new query $(e_1, r_n, ?a)$ or $(e_n, r_n, ?a)$ to the respective logical forms, ensuring that the intersection of the answers obtained from the new queries with those from $l$ is non-empty. Through this extension, both the path-like query and tree-like query are converted into the tree-like structures.

For both a path-like query (where $n \geq 2$) and a tree-like query, the two intersection extensions are applied with equal probability. In contrast, for single-triple queries (a special case of path-like queries), four operations are equally likely: union extension, two types of intersection extension, and no extension (to preserve some single-triple queries). The extension process results in four query types: *single-tree*, *union-tree-like*, *tree-like*, and *path-like*, corresponding to the attribution complexity types (denoted by $r$)—*single*, *union*, *intersection*, and *concatenation*.

## 4.3 STRUCTURED ATTRIBUTION GENERATION

We first obtain a KG subgraph $\mathcal{G}$ by grounding each extended query $l'$ in the KG, which returns the entities that are assigned to all the variables in the query for inferring the answer. The subgraph $\mathcal{G}$ is regarded as the structured attribution to support the answer to the question and falls under the *supportive* attribution category. To get structured attributions of the other three categories, i.e., *partially supportive*, *contradictory*, and *irrelevant*, we apply the following strategies to edit $\mathcal{G}$.

---

[2]Our methods can easily extend to more complex attribution cases using advanced logical operations like Negation and Kleene Plus (+) (Ren et al., 2023), which we leave for future exploration.

- **Partially Supportive**. The *partially supportive* subgraph $\mathcal{G}_{In}$ is generated by partial deletion, resulting in a subgraph that cannot fully support the answer. For path-like queries, we randomly delete one triple in $\mathcal{G}$. For tree-like or union-tree queries, we delete a path connecting one of the subject entities to the answer. In the case of single-triple queries, no deletion is performed.

- **Contradictory** The *contradictory* subgraph $\mathcal{G}_C$ is constructed by altering $\mathcal{G}$ such that its reasoning conflicts with the answer. This is done by replacing the answer entity in $\mathcal{G}$ with a non-answer entity of the same type. Especially for queries involving a union operation, we replace one of the answer entities within $\mathcal{G}$.

- **Irrelevant** The *irrelevant* subgraph $\mathcal{G}_{Ir}$ is obtained by selecting an entirely different subgraph from the KG that is structurally similar to $\mathcal{G}$ but contains unrelated entities and relations, except for the subject entity in $\mathcal{G}$.

### 4.4 DATA GENERATION

We employ GPT-3.5-turbo with tailored prompts to transform the subgraphs of $\mathcal{G}$, $\mathcal{G}_{In}$, $\mathcal{G}_C$ and $\mathcal{G}_{Ir}$ into natural language citations corresponding to the categories *supportive*, *partially supportive*, *contradictory* and *irrelevant*, respectively. When the original logical query $l$ is expanded to $l'$, the initial question $q$ is similarly extended to a new question $\tilde{q}$ using GPT-3.5-turbo. In addition, the answer entity $a$ is paraphrased into a more detailed answer statement $\tilde{a}$. Ultimately, this process yields an attribution QA sample consisting of the question $q$ or $\tilde{q}$, the answer statement $\tilde{a}$, the textual citation $c$, the attribution category $t$, and the reasoning complexity $r$. Further details on the generation process can be found in Appendix A.

## 5 EXPERIMENTAL SETUP

### 5.1 BENCHMARKS

**CAQA** Our CAQA benchmark is constructed following the method outlined in Section 4, combining two KGQA datasets: GrailQA (Gu et al., 2021) and WebQuestionsSP (Yih et al., 2016), along with the Freebase knowledge graph (Bollacker et al., 2008). CAQA consists of 161,174 samples, divided into a training set of 137,211 samples, which is used when the LLM needs fine-tuning or training, and a test set with 23,963 samples. Table 5 presents the distribution of these samples across different attribution categories and attribution complexity levels. Additionally, we manually annotated the attribution

Table 5: CAQA statistics across different attribution categories and different attribution complexity levels.

| Classes | | Train | Test | Total |
|---|---|---|---|---|
| | | 137,211 | 23,963 | 161,174 |
| Category | Sup. | 39,489 | 6,668 | 46,157 |
| | Ins. | 28,868 | 5,065 | 33,933 |
| | Con. | 36,620 | 6,423 | 43,043 |
| | Irr. | 32,234 | 5,807 | 38,041 |
| Complexity | S. | 73,795 | 10,443 | 84,238 |
| | C. | 46,783 | 8,455 | 55,238 |
| | U. | 5,347 | 886 | 6,233 |
| | I. | 11,286 | 4,179 | 15,465 |

categories of 300 test samples to assess their consistency with the automatically generated categories (see results in Section 6.2). Further details on CAQA's construction and statistics are provided in Appendix B, and human annotation processes are described in Appendix H.

**ALCE-FineGrained** We manually annotated 215 samples of the ALCE attributed QA benchmark according to the four attribution categories we proposed. The new benchmark, ALCE-FineGrained, is considered as an out-of-distribution (OOD) benchmark for comparing the performance of the attribution evaluator trained by our CAQA benchmark against existing specially developed automatic attribution evaluators. Additionally, we explore on this benchmark how attribution evaluators can be cost-effectively applied to OOD scenarios. Details of human annotation are given in Appendix H.

### 5.2 ATTRIBUTION EVALUATORS AND METRICS

We evaluate the LLM attribution evaluators in three settings: the zero-shot setting where the LLM is given none of the attribution samples; few-shot setting where the LLM is given a few attribution examples; and the fine-tuning setting where the LLM is trained with the samples in the training set. The LLMs of LLaMA-2 (Touvron et al., 2023b), LLaMA-3 (AI@Meta, 2024), Vicuna (Chiang

et al., 2023), and Mistral (Jiang et al., 2023) are tested for all the settings, with their different scales. LLaMA-3-70B, ChatGPT (`gpt-3.5-turbo-0613`) and GPT-4 (`gpt-4-0613`) are tested for the zero-shot and few-shot settings. Additionally, we test two specially developed automatic attribution evaluators AUTOIS (Honovich et al., 2022) and ATTRSCORE (Yue et al., 2023). More details on the implementation of the experiments are given in Appendix C.

In this work, we report the F1 score for the performance on each attribution category and the micro-F1 score for the performance on each complexity level and overall performance. Additionally, we include the FACTSCORES metric (Min et al., 2023) for a fine-grained evaluation of the "partially supportive" scenario (Section 6.3).

# 6 EXPERIMENTS

## 6.1 OVERALL RESULTS ON CAQA

Table 6: The performance of the different attribution evaluators on our CAQA benchmark. "-" indicates that it does not exist or is not applicable for comparison with others.

| Settings | Evaluators (Size) | Category | | | | | Complexity | | | |
|---|---|---|---|---|---|---|---|---|---|---|
| | | Sup. | Par. | Con. | Irr. | Overall | S. | C. | I. | U. |
| **Zero-Shot** | LLaMA-2 (7B) | 0.423 | 0.121 | 0.057 | 0.170 | 0.279 | 0.286 | 0.249 | 0.282 | 0.260 |
| | LLaMA-2 (13B) | 0.418 | 0.164 | 0.161 | 0.125 | 0.279 | 0.314 | 0.270 | 0.303 | 0.253 |
| | LLaMA-3 (8B) | 0.467 | 0.120 | 0.072 | 0.007 | 0.296 | 0.304 | 0.271 | 0.283 | 0.259 |
| | Mistral (7B) | 0.456 | 0.178 | 0.191 | 0.153 | 0.305 | 0.315 | 0.281 | 0.294 | 0.265 |
| | Vicuna (7B) | 0.513 | 0.100 | 0.064 | 0.199 | 0.327 | 0.343 | 0.273 | 0.312 | 0.256 |
| | Vicuna (13B) | 0.634 | 0.211 | 0.393 | 0.275 | 0.405 | 0.432 | 0.314 | 0.361 | 0.374 |
| | LLaMA-3 (70B) | 0.746 | 0.104 | 0.653 | 0.592 | 0.525 | 0.645 | 0.279 | 0.305 | 0.578 |
| | GPT-3.5-turbo | 0.583 | 0.017 | 0.598 | 0.512 | 0.497 | 0.555 | 0.321 | 0.363 | 0.363 |
| | GPT-4 | **0.771** | **0.456** | **0.745** | **0.473** | **0.630** | **0.685** | **0.451** | **0.514** | **0.616** |
| **Few-Shot** | LLaMA-2 (7B) | 0.300 | 0.066 | 0.009 | 0.334 | 0.248 | 0.259 | 0.218 | 0.167 | 0.308 |
| | LLaMA-2 (13B) | 0.419 | 0.199 | 0.167 | 0.089 | 0.272 | 0.274 | 0.271 | 0.233 | 0.267 |
| | LLaMA-3 (8B) | 0.573 | 0.202 | 0.234 | 0.156 | 0.336 | 0.356 | 0.279 | 0.310 | 0.294 |
| | Mistral (7B) | 0.412 | 0.152 | 0.041 | 0.415 | 0.349 | 0.339 | 0.278 | 0.300 | 0.271 |
| | Vicuna (7B) | 0.578 | 0.183 | 0.081 | 0.324 | 0.325 | 0.337 | 0.272 | 0.354 | 0.311 |
| | Vicuna (13B) | 0.633 | 0.208 | 0.383 | 0.288 | 0.403 | 0.427 | 0.315 | 0.397 | 0.374 |
| | LLaMA-3 (70B) | 0.741 | 0.182 | 0.608 | 0.584 | 0.521 | 0.628 | 0.295 | 0.314 | **0.563** |
| | GPT-3.5-turbo | 0.602 | 0.031 | 0.340 | 0.604 | 0.467 | 0.512 | 0.324 | 0.384 | 0.368 |
| | GPT-4 | **0.794** | **0.520** | **0.728** | **0.653** | **0.680** | **0.745** | **0.492** | **0.473** | 0.559 |
| **Fine-Tuing** | LLaMA-2 (7B) | 0.922 | 0.897 | **0.944** | **0.933** | 0.926 | 0.923 | 0.815 | 0.931 | 0.921 |
| | LLaMA-2 (13B) | 0.929 | 0.907 | 0.938 | 0.923 | 0.925 | 0.954 | 0.824 | **0.936** | 0.939 |
| | LLaMA-3 (8B) | 0.935 | 0.901 | 0.935 | 0.928 | 0.926 | 0.935 | 0.820 | 0.930 | 0.924 |
| | Mistral (7B) | 0.927 | 0.908 | **0.944** | 0.849 | 0.882 | 0.935 | 0.831 | 0.921 | 0.905 |
| | Vicuna (7B) | 0.937 | 0.907 | 0.940 | 0.906 | 0.932 | **0.956** | 0.823 | **0.936** | 0.939 |
| | Vicuna (13B) | **0.942** | **0.923** | 0.939 | 0.923 | **0.933** | 0.950 | **0.847** | 0.935 | **0.940** |
| | AUTOIS (11B) | 0.609 | - | - | - | - | - | - | - | - |
| | ATTRSCORE (13B) | 0.687 | - | 0.523 | 0.541 | 0.521 | 0.559 | 0.410 | 0.432 | 0.353 |

Table 6 shows the results of the attribution evaluators on CAQA. Our analysis is as follows:

**All evaluators perform poorly in identifying fine-grained negative attribution categories, especially *partially supportive*, compared to *supportive* under the zero-shot setting.** In the zero-shot setting, all evaluators perform significantly lower on the three negative categories than on *supportive*, except for GPT-3.5-turbo, which performs slightly better on *contradictory* than on *supportive*. Smaller LLMs ($\leq$ 13B) perform extremely poorly on all three negative categories, suggesting that none of them are capable of distinguishing subtle differences between negative attributions, with only Vicuna-13B performing slightly better. In particular, the evaluator is weakest at identifying *partially supportive*, and this becomes more apparent as the model scale increases. GPT-3.5-turbo barely recognises *partially supportive* whereas the best performer, GPT-4, only scores 0.430. We find that evaluators often classify *partially supportive* as *supportive*, even though it is apparent that part of the information is missing. Additionally, models (e.g. LLaMA-2, LLaMA-3 and Mistral) with the instruction fine-tuning version do not necessarily outperform their original versions, although we give them clear definitions for each attribution category, which illustrates the limitation of current instruction data. Appendix D shows the full results.

**Fine-tuning is effective in improving the performance of attribution evaluators, whereas the few-shot prompt tends to introduce bias.** Fine-tuning with our training set significantly enhances the evaluators' performance, with most exceeding an F1 score of 90% across all the categories. This improvement underscores the effectiveness of fine-tuning, with Vicuna in particular performing best after fine-tuning. In addition, the attribution evaluators AutoIS and AttrScore, which are fine-tuned on other benchmarks, also demonstrated competitive performance with GPT-3.5-turbo. These results indicate that while LLMs face challenges in attribution evaluation, targeted tuning can markedly boost their abilities. In contrast, the few-shot prompt is not an effective way to improve attribution evaluators, and it only shows noticeable gains on the powerful GPT-4, weakening the performance of most other models. We find the few-shot prompt introduces new biases, e.g., GPT-3.5-turbo has scores of 59.8% and 51.2% on the *contradictory* and *irrelevant* categories in the zero-shot setting, whereas in the few-shot setting the corresponding scores become 34.0% and 60.4%. Additionally, we explore more few-shot settings in Appendix D.

**Evaluation on the attribution is often biased towards keyword co-occurrence between answers and citations, failing to capture the logical reasoning, especially with complex citations.** This bias is a primary reason why all the evaluators perform worse on more complex cases with e.g., *concatenation*, *intersection*, and *union*. Smaller LLM evaluators are particularly affected due to their limited logical reasoning capabilities. This issue persists even in the simpler *single* scenario. For example, consider a sample of the category of *irrelevant*: the question is "What is the soundtrack of the video game X?" The answer is, "The video game X's soundtrack is Y," and the evidence is, "Z is a video game designer who has designed games such as X." Here, the evaluator incorrectly treats attribution as *supportive* due to the co-occurring keywords "video game" and "X", neglecting the logic of the relation "Soundtrack_Of" in the answer. In contrast, GPT-4 performs the best because it can capture some logical relationships. This capability is evident in its better performance in identifying logical relationships in the *contradictory* category and recognizing more *partially supportive* cases. These tasks require capturing the relational facts from the evidence text and doing reasoning with them for the answer. However, for the attribution complexity levels of *concatenation* and *intersection*, which require complex logical reasoning and the integration of multiple citations, all evaluators perform poorly. This suggests the need for improved logical reasoning abilities in evaluators. Notably, in the fine-tuning setting, evaluators show significant improvement across all attribution complexities. However, more future work is required to study whether this improvement results from enhanced reasoning abilities or merely from learning the internal patterns of the data.

## 6.2 EVALUATION OF CONSISTENCY WITH HUMAN ANNOTATIONS

**Consistency on evaluating evaluators.** We assess the consistency between the categories generated by our method and those annotated by humans by treating both sets as ground truth. This allows us to compute the overall micro-F1 scores for the 17 evaluators on the CAQA dataset, as shown in Figure 2. The results demonstrate that the performance of different evaluators across the various category generation methods is basically comparable. Furthermore, the Pearson correlation coefficient between the two sets of overall results is 0.97, indicating a remarkably high level of agreement between the automatically generated and manually annotated categories. This confirms that evaluations based on automatically generated categories closely align with manual evaluations.

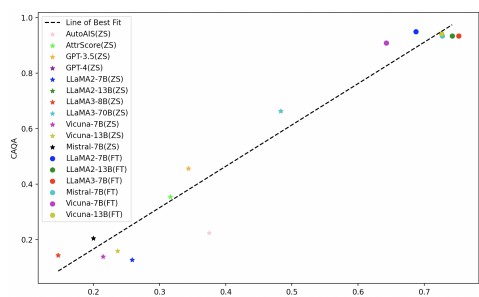

Figure 2: Correlation of (1) overall results of evaluators on CAQA based on the automatically generated categories (y-axis), and (2) overall results of evaluators on CAQA based on human-annotated categories (x-axis).

## 6.3 FINE-GRAINED EVALUATION IN THE PARTIALLY SUPPORTIVE SCENARIO

Our CAQA benchmark provides a more detailed evaluation compared to existing benchmarks, particularly in identifying when an attribution category is "partially supportive". Specifically, it quantifies how many sub-facts in an answer are supported by citations. The CAQA benchmark can automatically obtain the proportion of supported sub-facts without manual labeling. It does so by

calculating the difference in the number of triples between the initial subgraph and the subgraph after a deletion operation. We refer to FACTSCORES (Min et al., 2023) to further evaluate representative evaluators in the overall results. In our approach, we first convert the triples in the initial subgraph $\mathcal{G}$ into natural language sub-facts using ChatGPT. Then, FACTSCORES metrics are applied to all evaluators, indicating the proportion of sub-facts in the answers that are supported by citations. Additional implementation details are provided in Appendix C.

The experimental results presented in Table 7 reveal a significant performance gap between current evaluators and human evaluators in fine-grained attribution assessment. Notably, evaluators that identify more attribution categories perform worse. For example, the three evaluators fine-tuned on the CAQA benchmark, which can identify four attribution categories, and AttrScore, which identifies three, exhibit much higher error rates compared to AutoIS, which identifies only two categories. In contrast, evaluators in the zero-shot setting tend to overestimate FACTSCORES, as their attribution assessments are biased by keyword co-occurrence in sub-facts and citations—consistent with the findings in Section 6.1. Additionally, the FACTSCORES of the automated annotations generated by our CAQA benchmark differ from human annotations by only 4%, demonstrating that the CAQA benchmark provides a reliable framework for automated fine-grained evaluation.

Table 7: Performance of representative evaluators on 200 partially supportive samples. FActScore (FS) indicates the proportion of subfacts supported by citations, while Error Rate (ER) measures the discrepancy between the evaluator's results and Human evaluation. CAQA* refers to the annotations automatically generated by our benchmark. **Bold** indicates the best (lowest) ER.

|  | Evaluators | FS | ER |
|---|---|---|---|
| **Zero-Shot** | LLaMA-3 (70B) | 0.85 | 0.27 |
|  | GPT-3.5-turbo | 0.93 | 0.35 |
|  | GPT-4 | 0.84 | 0.26 |
| **Fine-Tuning** | LLaMA-3 (8B) | 0.19 | 0.39 |
|  | Vicuna (7B) | 0.19 | 0.39 |
|  | Vicuna (13B) | 0.18 | 0.40 |
|  | AUTOIS (11B) | 0.44 | 0.14 |
|  | ATTRSCORE (13B) | 0.25 | 0.33 |
|  | CAQA* | 0.62 | **0.04** |
|  | Human | 0.58 | - |

## 6.4 EXPLORATION OF OUT-OF-DOMAIN DATA

We test the baselines AutoIS (based on T5-11B) and AttrScore (based on Vicuna-13B) that are trained by some other benchmarks, and T5-11B and Vicuna-13B fine-tuned by CAQA, on the OOD benchmark ALCE-FineGrained. For comparison with AutoIS, we merge the three negative categories into Non-Supportive. The results are shown in Table 8. Compared to AutoIS and AttrScore, T5-11B* and Vicuna-13B*, fine-tuned by CAQA, have competitive performance in individual classes and the overall score. This demonstrates that CAQA is more effective for developing attribution evaluators using the existing LLMs. Table 8 also verifies that fine-tuning with a few samples of the domain of the testing samples is effective in improving the evaluators. Further details can be found in Appendix E.

Table 8: Performance of (1) T5-11B* and Vicuna-13B* (LLMs fine-tuned by CAQA) and (2) AutoIS and AttrScore, when tested on ALCE-FineGrained.

| Evaluators | ALCE-FineGrained | | | | |
|---|---|---|---|---|---|
|  | Sup. | | Non-Sup. | | Overall |
| AutoIS (T5-11B) | 0.31 | | 0.65 | | 0.54 |
| T5-11B* | **0.44** | | **0.72** | | **0.63** |
|  | Sup. | Par. | Con. | Irr. | Overall |
| AttrScore (Vicuna-13B) | 0.52 | - | 0.21 | 0.42 | 0.36 |
| Vicuna-13B* | 0.54 | 0.24 | 0.30 | 0.34 | 0.38 |
| Vicuna-13B* Few-Shot | 0.51 | 0.29 | 0.16 | 0.34 | 0.36 |
| Vicuna-13B* Fine-Tuning | **0.69** | **0.36** | **0.40** | **0.46** | **0.52** |

## 7 CONCLUSION AND FUTURE WORK

This work has advanced the field of analyzing and developing evaluators for natural language QA attribution in the era of LLM. To this end, we presented a comprehensive set of attribution criteria and developed an automatic approach that can construct attributed QA benchmarks with complete and fine-grained attribution categories and different attribution complexity levels using KGs. We have not only analyzed multiple LLM-based automatic evaluators and verified the effectiveness of the generated benchmark CAQA, but also compared the automatically generated categories with human annotated categories, showing their high consistency. Our findings reveal that while current evaluators generally struggle with attribution, targeted tuning can significantly improve their capabilities. This advancement holds promise for refining LLM performance, particularly in addressing factuality and faithfulness hallucination issues. In the future, we will study using CAQA and its other versions to augment QA attributions by providing evaluation feedback.

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

# A  GENERATION OF NATURAL LANGUAGE QUESTIONS, ANSWERS AND ATTRIBUTIONS

This section presents examples for generating natural language questions, answers, and citations using the GPT-3.5-turbo model. Our approach involves using distinct prompts for transforming subgraphs into comprehensible natural language citations, extending original questions, and converting answer entities into detailed answer statements. Table 9 demonstrates the conversion of knowledge graph subgraphs into natural language citations. Table 10 illustrates the example of generating the extended question. Table 11 provides an example of how answer entities are transformed into long-form answer statements.

---

**Instruction:** Your task is to convert a specific subgraph from the knowledge graph into one or more coherent sentences that summarize the information encapsulated within the subgraph.

**Subgraph**: [("Wii/DS NA", "type", "computer_game_region"), ("LostWinds: Winter of the Melodias", "type", "game_version"), ("LostWinds: Winter of the Melodias", "game_version.regions", "Wii/DS NA"), ("LostWinds: Winter of the Melodias", "game_version.distributed_through", "WiiWare"), ("Frontier Developments", "cvg_developer.game_versions_developed", "LostWinds: Winter of the Melodias")].

**Sentences:** LostWinds: Winter of the Melodias is a computer game with a specific version tailored for the Wii/DS NA region. This version is distributed through WiiWare. The game was developed by Frontier Developments, showcasing their involvement in creating various game versions.

---

Table 9: An example about converting a subgraph to a natural language citation using GPT-3.5-turbo.

---

**Instruction:** Given knowledge graph triples, your task is to generate a question using all the triples. The generated questions should contain all the relationships.

# Extended Triples
**Triples:** [(?x, type, cvg.computer_videogame), (?x, computer_videogame.influenced_by, Sengoku Rance), (?x, fictional.setting, Touhou Project)]

**Question question:** What computer video game was influenced by Sengoku Rance and is set in the Touhou Project fictional universe?

---

Table 10: An example about generating the extended question using GPT-3.5-turbo.

---

**Instruction:** Your task is to convert a question along with its concise answer into a comprehensive answer statement.

**Question:** What group fought in the Battle of Vicksburg that was based in Montgomery?
**Answer:** Army of Mississippi

**Answer statement:** The group that fought in the Battle of Vicksburg and was based in Montgomery was the Army of Mississippi.

---

Table 11: An example about converting the answer entity to a long answer statement using GPT-3.5-turbo.

# B  CAQA BENCHMARK CONSTRUCTION AND STATISTICS

The CAQA benchmark is built on the top of two KGQA datasets, GrailQA and WebQuestionsSP, with the knowledge graph Freebase, forming a comprehensive attribution evaluation testbed. We selectively include samples from these two datasets whose logical queries align with single-triple,

path-like, or tree-like queries, as delineated in Section 4.1. For path queries, we collect the example with a path length of at most two hops. We treat paths incorporating CVT (Compound Value Type) nodes as one-hop. For example, [*(Harper Lee, person.education ?cvt), (?cvt education.institution, Monroe County High School)*] is a one-hop path, where the node *?cvt* holds no actual meaning. Regarding tree-liked queries, we restrict our selection to those with a maximum of two non-answer nodes, meaning up to two subject entities.

The length distribution (i.e., the number of tokens) of citations in the training and test sets of the CAQA benchmark is depicted in Figures 3 and 4. These distributions reveal a concentration of citations around 25 tokens, with a minority exceeding 60 tokens. In future work, we aim to enhance the complexity and length of natural language references by developing more intricate subgraphs. Additionally, Figure 5 presents the domain distribution within the CAQA benchmark. This distribution underscores the benchmark's broad domain coverage and its encompassment of various sub-domains, highlighting the diversity of our benchmark.

## C  IMPLEMENTATION DETAILS

Table 12 describes the different prompt designs against the various attribution evaluators. AutoIS is a natural language inference (NLI) model[3] based on T5-11B that outputs a "1" to indicate that the citation supports the answer statement or a "0" to indicate a lack of support. AttrScore is a uniform name for attribution evaluators developed on various LLMs, and we use the best-performing attribution evaluator (Vicuna-13B) on the original work for comparison. Since AutoIS can only recognise *supportive* and *non-supportive* attribution categories, we only report its F1 score on *supportive* in Table 6. In the experiments on the ALCE-FineGrained benchmark, to be able to compare the evaluator trained on our benchmark with AutoIS, we merge the three incorrect categories into the *non-supportive* category, and then compute F1 scores of *supportive* and *non-supportive* as well as overall micro-F1 score.

In the few-shot setting, we select one sample per attribution category as a demonstration, as shown in Table 13. We explore on more few-shot settings in Appendix D. For model fine-tuning, we use the prompt of "Other Evaluators" depicted in Table 12 as input of all models, and the output of the model is one of the four attribution categories proposed. We use two A100 80G GPUs for full parameter fine-tuning and one A100 80G GPU for the inference phase. During inference, text generation is conducted with a temperature setting of 0. If LLMs produce an attribution category with an explanation, we extract the predicted label using regular expression techniques.

For the fine-grained evaluation in the *partially supportive* scenario, we use GPT-3.5 to convert triples into natural language subfacts with the prompt: "Your task is to convert a triple into natural language statement". Following the Retrieve→LM method (Min et al., 2023), the prompt is fed into the evaluator, which predicts True or False. For the zero-shot evaluator, we use the prompt: "*Judge this fact based on the given context.\n\n Fact: {sub-fact}\n Text: {citation} \n\nTrue or False?\nOutput:*". For fine-tuned and existing evaluators, the prompt provided in Table 12 is used. When the evaluator incorporates more than two attribution categories, we categorize supportive as True and all other categories as False for calculating the FACTSCORES. Human annotation, as described in Appendix H, involves annotators determining whether each subfact is supported by its citation. The FACTSCORES is the proportion of predictions classified as True compared to the total number of subfacts evaluated.

## D  DETAILED EXPERIMENTAL RESULTS

| N-shot (GPT-3.5-turbo) | CAQA | | | | |
|---|---|---|---|---|---|
| | Sup. | Par. | Con. | Irr. | Overall |
| 1-shot | 0.613 | 0.026 | 0.318 | **0.609** | 0.476 |
| 2-shot | **0.627** | **0.034** | 0.359 | 0.593 | **0.486** |
| 3-shot | 0.599 | 0.015 | **0.378** | 0.581 | 0.478 |

Table 14: The performance of GPT-3.5-turbo under various few-shot settings on CAQA.

We present the full experimental results in Tables 15. Additionally, we investigate three few-shot settings: 1-shot, 2-shot, and 3-shot in 5,000 test instances employing GPT-3.5-turbo. In these settings, 1, 2, and 3 examples, respectively, are provided for each attribution category. The outcomes, as displayed in Table 14, suggest that increasing the number of examples yields negli-

[3]https://huggingface.co/google/t5_xxl_true_nli_mixture

**GPT-3.5 and GPT-4**
**Instruction:** Your task is to evaluate the relationship between a provided citation and the answer to a specific question. There are four possible types of relationships:
1. Supportive: Choose this if the citation directly confirms or is fully in alignment with the answer, providing all necessary information to substantiate it.
2. Insufficient: Choose this when the citation provides only partial backing for the answer, lacking some essential details or evidence needed for full support.
3. Contradictory: Choose this option if the citation is consistent with the intent of the question but directly opposes or contradicts the answer.
4. Irrelevant: Select this option if the citation does not match the intent of the question and contains information that is not useful for answering.
For each example provided: First, you need to look at the question given and the answer provided. Then, compare them with the content of the citation. Finally, select the appropriate relationship category based on whether the citation supports the answer, is missing information, contradicts itself, or is irrelevant to the answer.
Example:
**Question:** {question}
**Answer:** {answer statement}
**Reference:** {citation}
**Relationship Category:**

---

**AttrScore**
**premise:** {question|answer statement}
**hypothesis:** {citation}

---

**AutoIS**
Below is an instruction that describes a task, paired with an input that provides further context. Write a response that appropriately completes the request.
**Instruction:** Verify whether a given reference can support the claim. Options: Attributable, Extrapolatory or Contradictory.
**Claim:** {question|answer statement}
**Reference:** {citation}
**Response:**

---

**Other Evaluators**
Below is an instruction that describes a task, paired with an input that provides further context. Write a response that appropriately completes the request.
**Instruction:** Verify whether a given reference can support the claim. Options: Supportive, Insufficient, Contradictory or Irrelevant.
**Claim:** {question|answer statement}
**Reference:** {citation}
**Response:**

Table 12: Different prompts designed for different evaluators.

gible improvement in performance. Consequently, considering the associated costs, we have opted to use the 1-shot setting in all subsequent experiments.

# E    DETAILS OF EXPERIMENTS ON ALCE-FINEGRAINED

ALCE-FineGrained consists of 215 manually labelled samples containing 104 supportive samples, 58 partially supportive samples, 25 contradictory samples, and 28 irrelevant samples. For the few-shot setting, we select one sample for each attribution category as demonstration. For the fine-tuning setting, we employ GPT-4 to annotate 800 samples from the ALCE benchmark as the training set. Since there are fewer contradictory and irrelevant attribution categories in the ALCE benchmark, we use GPT-4 to edit the evidence to construct contradictory and irrelevant samples, thus ensuring a balanced number of the four categories.

Table 16 presents two ALCE-FineGrained examples, illustrating the attribution categories *partially supportive* and *irrelevant*, respectively. It shows that these two categories, which are not included in

**GPT-3.5 and GPT-4**
**Instruction:** Your task is to evaluate the relationship between a provided citation and the answer to a specific question. There are four possible types of relationships:
1. Supportive: Choose this if the citation directly confirms or is fully in alignment with the answer, providing all necessary information to substantiate it.
2. Insufficient: Choose this when the citation provides only partial backing for the answer, lacking some essential details or evidence needed for full support.
3. Contradictory: Choose this option if the citation is consistent with the intent of the question but directly opposes or contradicts the answer.
4. Irrelevant: Select this option if the citation does not match the intent of the question and contains information that is not useful for answering.
Please read the examples and choose the most appropriate relationship category for the test example.
Example 1: {Support Example}
Example 2: {Missing Example}
Example 3: {Contradictory Example}
Example 4: {Irrelevant Example}
Test Example:
**Question:** {question}
**Answer:** {answer statement}
**Reference:** {citation}
**Relationship Category:**

────────────────────────────────────────────────

**Other Evaluators**
Below is an instruction that describes a task, paired with an input that provides further context. Write a response that appropriately completes the request.
**Instruction:** Verify whether a given reference can support the claim. Options: Supportive, Insufficient, Contradictory or Irrelevant.
{Support Example}
{Missing Example}
{Contradictory Example}
{Irrelevant Example}
**Claim:** {question|answer statement}
**Reference:** {citation}
**Response:**

Table 13: Different few-shot prompts designed for different evaluators.

the previous attribution categories, are common and different in practical situations. In example 1, where the attribution category is *partially supportive*, most of the answer statement (highlighted in green) is mentioned in the citation, but the key information "*The Maryland Transportation Authority*" (highlighted in yellow) is not mentioned in the citation. This demonstrates the subtleties that can render an attribution insufficient. In example 2, which is categorised as *irrelevant*, the entirety of the answer statement is irrelevant to the citation. This exemplifies a clear case of irrelevant attribution. Notably, previous evaluators, AutoIS and AttrScore, are unable to accurately classify these cases. In contrast, Vicuna, an evaluator trained with our CAQA benchmark, successfully identifies the correct attribution categories. This underscores the effectiveness and practicality of employing the CAQA benchmark for developing attribution evaluators.

## F  ANALYSIS OF EXISTING ATTRIBUTED QA SYSTEMS

Following the work of Gao et al. (Gao et al., 2023b) we reproduce the attributed question answering system based on Vicuna-13B model, noted for its effectiveness in smaller language model configurations. Specifically, we provide the model with the top-3 retrieved passages and instruct the model to cite them accordingly. The retrieved passages and the instruction are consistent with the original implementation. Upon reviewing 234 instances of the system, our analysis revealed that: 44.4% of the instances accurately cited evidence supporting their answers, while 24.8% cited evidence that only partially supported the answers. Contradictory evidence was cited in 10.7% of cases, and 12.0% of the responses involved citations of irrelevant evidence. Additionally, 8.1% of the cases were categorized under other issues, including incomplete or unclear answers. The predominant challenges

| Settings | Evaluators (Size) | Category | | | | | Complexity | | | |
|---|---|---|---|---|---|---|---|---|---|---|
| | | Sup. | Par. | Con. | Irr. | Overall | S. | C. | I. | U. |
| **Zero-Shot** | LLaMA-2 (7B) | 0.423 | 0.121 | 0.057 | 0.170 | 0.279 | 0.286 | 0.249 | 0.282 | 0.260 |
| | LLaMA-2-chat (7B) | 0.462 | 0.158 | 0.058 | 0.053 | 0.183 | 0.281 | 0.235 | 0.291 | 0.290 |
| | LLaMA-2 (13B) | 0.418 | 0.164 | 0.161 | 0.125 | 0.279 | 0.314 | 0.270 | 0.303 | 0.253 |
| | LLaMA-2-chat (13B) | 0.469 | 0.171 | 0.173 | 0.103 | 0.224 | 0.338 | 0.279 | 0.305 | 0.278 |
| | LLaMA-3 (8B) | 0.467 | 0.120 | 0.072 | 0.007 | 0.296 | 0.304 | 0.271 | 0.283 | 0.259 |
| | LLaMA-3-Instruct (8B) | 0.492 | 0.166 | 0.178 | 0.131 | 0.314 | 0.312 | 0.285 | 0.295 | 0.289 |
| | Mistral (7B) | 0.456 | 0.178 | 0.191 | 0.153 | 0.305 | 0.315 | 0.281 | 0.294 | 0.265 |
| | Mistral-Instruct (7B) | 0.591 | 0.189 | 0.159 | 0.016 | 0.324 | 0.339 | 0.278 | 0.300 | 0.271 |
| | Vicuna (7B) | 0.513 | 0.100 | 0.064 | 0.199 | 0.327 | 0.343 | 0.273 | 0.312 | 0.256 |
| | Vicuna (13B) | 0.634 | 0.211 | 0.393 | 0.275 | 0.405 | 0.432 | 0.314 | 0.361 | 0.374 |
| | LLaMA-3 (70B) | 0.746 | 0.104 | 0.653 | 0.592 | 0.525 | 0.645 | 0.279 | 0.305 | 0.578 |
| | GPT-3.5-turbo | 0.583 | 0.017 | 0.598 | 0.512 | 0.497 | 0.555 | 0.321 | 0.363 | 0.363 |
| | GPT-4 | **0.771** | **0.456** | **0.745** | **0.473** | **0.630** | **0.685** | **0.451** | **0.514** | **0.616** |
| **Few-Shot** | LLaMA-2 (7B) | 0.300 | 0.066 | 0.009 | 0.334 | 0.248 | 0.259 | 0.218 | 0.167 | 0.308 |
| | LLaMA-2-chat (7B) | 0.281 | 0.008 | 0.005 | 0.364 | 0.219 | 0.281 | 0.235 | 0.291 | 0.290 |
| | LLaMA-2 (13B) | 0.419 | 0.199 | 0.167 | 0.089 | 0.272 | 0.274 | 0.271 | 0.233 | 0.267 |
| | LLaMA-2-chat (13B) | 0.424 | 0.185 | 0.125 | 0.114 | 0.273 | 0.338 | 0.279 | 0.305 | 0.278 |
| | LLaMA-3 (8B) | 0.573 | 0.202 | 0.234 | 0.156 | 0.336 | 0.356 | 0.279 | 0.310 | 0.294 |
| | LLaMA-3-Instruct (8B) | 0.593 | 0.197 | 0.365 | 0.272 | 0.398 | 0.356 | 0.279 | 0.310 | 0.294 |
| | Mistral (7B) | 0.552 | 0.152 | 0.041 | 0.415 | 0.349 | 0.339 | 0.278 | 0.300 | 0.271 |
| | Mistral-Instruct (7B) | 0.563 | 0.267 | 0.171 | 0.424 | 0.393 | 0.415 | 0.291 | 0.354 | 0.395 |
| | Vicuna (7B) | 0.578 | 0.183 | 0.081 | 0.324 | 0.325 | 0.337 | 0.272 | 0.354 | 0.311 |
| | Vicuna (13B) | 0.633 | 0.208 | 0.383 | 0.288 | 0.403 | 0.427 | 0.315 | 0.397 | 0.374 |
| | LLaMA-3 (70B) | 0.741 | 0.182 | 0.608 | 0.584 | 0.521 | 0.628 | 0.295 | 0.314 | **0.563** |
| | GPT-3.5-turbo | 0.602 | 0.031 | 0.340 | 0.604 | 0.467 | 0.512 | 0.324 | 0.384 | 0.368 |
| | GPT-4 | **0.794** | **0.520** | **0.728** | **0.653** | **0.680** | **0.745** | **0.492** | **0.473** | 0.559 |
| **Fine-Tuing** | LLaMA-2 (7B) | 0.922 | 0.897 | **0.944** | **0.933** | 0.926 | 0.923 | 0.815 | 0.931 | 0.921 |
| | LLaMA-2-chat (7B) | 0.925 | 0.903 | 0.943 | 0.927 | 0.930 | 0.935 | 0.820 | 0.930 | 0.924 |
| | LLaMA-2 (13B) | 0.929 | 0.907 | 0.938 | 0.923 | 0.925 | 0.954 | 0.824 | **0.936** | 0.939 |
| | LLaMA-2-chat (13B) | 0.931 | 0.902 | 0.939 | 0.927 | 0.926 | 0.953 | 0.825 | 0.934 | 0.939 |
| | LLaMA-3 (8B) | 0.935 | 0.901 | 0.935 | 0.928 | 0.926 | 0.935 | 0.820 | 0.930 | 0.924 |
| | Mistral (7B) | 0.927 | 0.908 | **0.944** | 0.849 | 0.882 | 0.935 | 0.831 | 0.921 | 0.905 |
| | Vicuna (7B) | 0.937 | 0.907 | 0.940 | 0.906 | 0.932 | **0.956** | 0.823 | **0.936** | 0.939 |
| | Vicuna (13B) | **0.942** | **0.923** | 0.939 | 0.923 | **0.933** | 0.950 | **0.847** | 0.935 | **0.940** |

Table 15: Full results on CAQA

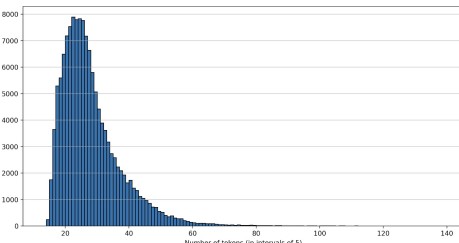

Figure 3: Histogram of the number of tokens across all citations in the CAQA benchmark training set.

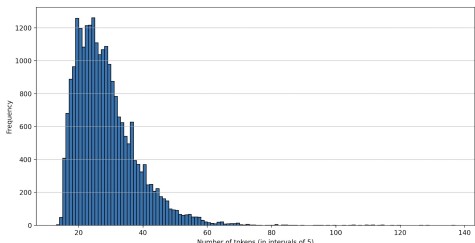

Figure 4: Histogram of the number of tokens across all citations in the CAQA benchmark test set.

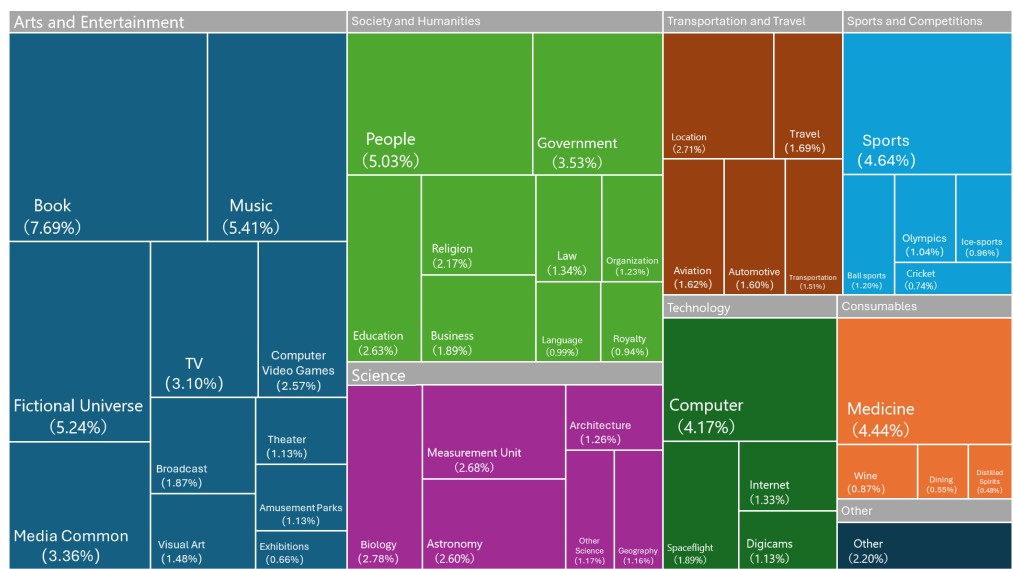

Figure 5: The distribution of examples across different domains in the CAQA benchmark.

in incorrect attributions are identified as *partially supportive*, *contradictory*, and *irrelevant* citations, with *partially supportive* citations being the most common problem.

## G LIMITATIONS

This work introduces a benchmark with a detailed attribution category and four attribution complexities based on distinct reasoning types. However, we recognize several limitations in the current design. Our benchmark does not address more intricate attribution scenarios that pose significant challenges. These include instances involving lengthy answers and citations, mathematical reasoning within attributions, and scenarios that necessitate commonsense knowledge for accurate interpretation.

For illustration, consider the question: "When did England last reach the quarterfinals of the World Cup?" The provided answer is "England last made the quarterfinals in 1990," with a citation noting that "The England national football team finished in fourth place in 2018 and reached the semifinals in 1990." To accurately attribute the answer, it is essential to understand that finishing in fourth place implies participation in the quarterfinals and that 2018 is more recent than 1990.

To address these shortcomings, our future work could include expanding the attribution graph to accommodate longer answers and citations, integrating numerical answers with mathematical reasoning, and developing common-sense knowledge graphs. These improvements may make our benchmarks more relevant to real-world challenges.

> **Example 1**
> **Question:** The Maryland Transportation Authority is in charge of what landmarks?
> **Answer Statement:** The Maryland Transportation Authority is in charge of providing funding for the construction of revenue-producing transportation facilities for the Maryland Department of Transportation, which have included improvements at the Port of Baltimore and the Baltimore-Washington International Airport
> **Citation:** of a commercial enterprise. Its capital projects and operations are funded by tolls, concessions, investment income, and revenue bonds. In addition to its own toll facilities, the Authority finances construction of other revenue-producing transportation facilities for the Maryland Department of Transportation (MDOT). These projects have included improvements at the Port of Baltimore and the Baltimore-Washington International Airport. To provide construction funding, the Authority issues revenue bonds, which will be paid off over a period of years by tolls and other user fees generated by the facilities. The MDTA can issue either taxable or exempt bonds to finance large scale projects.
> **AutoIS:** Supportive ✗
> **AttrScore:** Irrelevant ✗
> **Vicuna[†]:** Partially Supportive ✓
>
> ──────────────────────────────────────────────
>
> **Example 2**
> **Question:** When did the last season of jersey shore air?
> **Answer Statement:** The TV show Jersey Shore aired its final episode on December 20, 2012.
> **Citation:** 8.56 million viewers, only to set another record with the airing of the fourth episode, which garnered 8.87 million viewers. On January 25, 2011, it was confirmed that the show had been renewed for a fourth season, to be filmed in Italy during the first half of 2011. The fourth season premiered August 4, 2011. MTV confirmed in June 2011 that the fifth season would return to Seaside Heights. Believed complications caused by Nicole Polizzi's pregnancy, and several cast members (including Polizzi, DelVecchio, and Farley) receiving spin-offs sparked talk about the future of the series past the fifth season, however
> **AutoIS:** Supportive ✗
> **AttrScore:** Contradictory ✗
> **Vicuna[†]:** Irrelevant ✓

Table 16: Two examples of the results of the three attribution evaluators on ALCE-FineGrained. Content in yellow highlights portions of the answer statement not found in the citation, whereas green indicates content present in the citation.

# H    HUMAN ANNOTATION

The human annotation process for our study was conducted by the authors themselves, eliminating the need for external paid services. Three of our annotators were asked to read these guidelines carefully. Only annotators with a thorough understanding of the guidelines and the task were allowed to participate in the manual evaluation. We ensured the reliability of the results by retaining only those annotations that were aligned across all three annotators. Annotation guidelines are shown in Fig. 6 and 7.

You will see a question, the corresponding answer, and the cited reference. What you need to do is:

1. Read the question, the answer and the cited reference carefully.

2. You should judge whether the cited reference is *supportive, partially supportive, contradictory,* or *irrelevant* to answer of the question.

· **Supportive**: The cited reference includes facts that can fully support the answer.

· **Partially Supportive**: The cited reference lacks a part of the facts that are required to infer the answer.

· **Contradictory**: The cited reference includes facts that can infer a different answer.

· **Irrelevant**: The cited reference has no facts that can be used to infer the answer.

Here are some examples of the four categories:

**1. Supportive**

| |
|---|
| **Question:** Who is hosting the next world cup 2022? |
| **Answer:** The 2022 FIFA World Cup will be hosted by Qatar |
| **Reference:** Title: 2018 and 2022 FIFA World Cup bids. Content: FIFA's headquarters in Zurich. Russia was chosen to host the 2018 World Cup, and Qatar was chosen to host the 2022 World Cup. This made Russia the first Eastern European country to host the World Cup, while Qatar would be the first Middle Eastern country to host the World Cup. Blatter noted that the committee had decided to "go to new lands" and reflected a desire to "develop football" by bringing it to more countries. In each round a majority of twelve votes was needed. If no bid received 12 votes in a round, the bid with the fewest votes |
| **Question:** Who lived to be the oldest person in the world? |
| **Answer:** The longest-lived human on record was Jeanne Calment, who lived to be 122 years and 164 days old |
| **Reference:** Title: Oldest people. Content: Oldest people This is a list of tables of the oldest people in the world in ordinal ranks. To avoid including false or unconfirmed claims of extreme old age, names here are restricted to those people whose ages have been validated by an international body that specifically deals in longevity research, such as the Gerontology Research Group (GRG) or "Guinness World Records" (GWR), and others who have otherwise been . According to this criterion, the longest human lifespan is that of Jeanne Calment of France (1875–1997), who lived to the age of 122 years, 164 days. She met Vincent van |

**2. Partially Supportive**

| |
|---|
| **Question:** What do you use to test for lipids? |
| **Answer:** To test for lipids, a blood sample is taken after a 12-hour fast, which is then used to measure a lipid profile through mass spectrometry, chromatography, or nuclear magnetic resonance |
| **Reference:** Title: Cholesterol. Content: and then every 3–12 months thereafter. A blood sample after 12-hour fasting is taken by a doctor, or a home cholesterol-monitoring device is used to measure a lipid profile, an approach used to estimate a person's lipoproteins, the vastly more important issue because lipoproteins have always been concordant with outcomes though the lipid profile is commonly discordant LDL Particle Number and Risk of Future Cardiovascular Disease in the Framingham Offspring Study. The lipid profile measures: (a) total cholesterol, (b) cholesterol associated with HDL (i.e. Higher Density {than water} Lipids-transported-within-proteins) particles ("which can regress arterial disease"), (c) triglycerides and (d) (by |

Figure 6: First page of the annotation guidelines.

**Question:** Where was in the dark tv series filmed?

**Answer:** In the Dark, a British crime drama series, was filmed in Manchester and Marsden

**Reference:** Title: In the Dark (UK TV series). Content: of kidnapping two young girls. In the second two-parter, a heavily pregnant Helen is pulled into the dark side of urban Manchester as she deals with an unexpected tragedy. Filming for the series began in April 2017 in Manchester and Marsden. "The Daily Telegraph"s Michael Hogan gave the first episode three stars out of five, noting that: ""In the Dark" did show promise and could yet come good. It was taut and tensely atmospheric with an intriguing premise which found its heroine caught in the middle between police and prime suspect." Reviewing the first episode, "The Guardian"s Sam Wollaston concluded

### 3. Contradictory

**Question:** When did spain win their first world cup?

**Answer:** Spain won their first FIFA World Cup in 1964, hosted in their home country

**Reference:** Title: Spain national football team. Content: the silver medal. Spain qualified for their first FIFA World Cup in 1934, defeating Brazil in their first game and losing in a replay to the hosts and eventual champions Italy in the quarter-finals. The Spanish Civil War and World War II prevented Spain from playing any competitive matches between the 1934 World Cup and the 1950 edition's qualifiers. At the 1950 finals in Brazil, they topped their group to progress to the final round, then finished in fourth place. Until 2010, this had been Spain's highest finish in a FIFA World Cup finals, which had given them the name

**Question:** Who was the last person hanged in england?

**Answer:** Peter Manuel was the last person to be hanged in the UK for killing a police officer

**Reference:** Title: Harry Allen (executioner). Content: 1957 reduced the number of condemned criminals by 75%, from an average of 15 a year in the early 1950s to about four a year in the late 1950s. As Chief Executioner, on 11 July 1958 Allen hanged American-born Scottish serial killer Peter Manuel at Barlinnie prison, Glasgow. He also hanged Guenther Podola on 5 November 1959, a German-born petty thief, and the last man to be hanged in the UK for killing a police officer. His most controversial case was that of James Hanratty, hanged on 4 April 1962 at Bedford Prison for the "A6 murder" case. Efforts to

### 4. Irrelevant

**Question:** Who plays patrick in 10 things i hate about you?

**Answer:** Patrick is played by actor Heath Ledger in the 1999 film 10 Things I Hate About You

**Reference:** Title:10 Things I Hate About You. Content: assists by convincing Joey to pay Patrick to take out Kat, under the pretense that this will allow Joey to date Bianca. Patrick agrees to the deal, but Kat rebuffs his first few advances. Michael and Cameron help him by prying Bianca for information on Kat's likes and dislikes. Armed with this knowledge, Patrick begins to win Kat's interest. She goes to a party with him, which enables Bianca to go as well, much to Walter's dismay. At the party, Kat becomes upset when she sees Bianca with Joey,

**Question:** How many medals did australia win in the 2000 olympics?

**Answer:** According to the information provided in the search results, Australia won a total of 58 medals at the 2000 Summer Olympics, with 14 gold, 26 silver, and 28 bronze

**Reference:** Title: 2000 Summer Paralympics medal table. Content: The location and facilities were shared with the largest event, the 2000 Summer Olympics, which concluded on 1 October. The Games set records for athlete and country participation, tickets sold, hits to the official Games website, and medals on offer. A record of 122 countries (or 123 delegations including independent athletes from Timor-Leste) participated; 68 countries won medals, of which seven won a medal for the first time. A total of 1,657 medals were awarded during the Sydney games: 550 gold, 549 silver, and 558 bronze. Among these performances,

Figure 7: Second page of the annotation guidelines.

