# OpenReview forum: "Can LLMs Evaluate Complex Attribution in QA? Automatic Benchmarking Using Knowledge Graphs"
_ICLR.cc/2025/Conference — Submitted to ICLR 2025_

### Official Review · Reviewer_yhfY · 2024-11-03

**Soundness:** 3
**Presentation:** 3
**Contribution:** 3
**Rating:** 6
**Confidence:** 3

**Summary:**

This manuscript aims to bridge the gap in attribution evaluators' use of knowledge graphs by providing detailed categories. This study's experiment offers multiple configurations in zero-shot, few-shot, and fine-tuned contexts, demonstrating that the fine-tuning process can significantly improve performance. This benchmark aims to address the shortcomings of existing attribution evaluators, which face challenges with intricate attribution categories and sophisticated reasoning processes.

**Strengths:**

The use of KGs for automatic construction of the benchmark is novel, making the process scalable and adaptable.
The research tests various models and demonstrates the needs of fine-tuning for achieving robust performance.
The benchmark shows high consistency with human evaluations, supporting its credibility as an effective tool for future developments in QA systems.
The choice to test multiple LLMs, including state-of-the-art models like GPT-4 and LLaMA variants, provides a robust analysis of performance across different model scales and settings.
The inclusion of a wide range of attribution types and complexity levels sets a high standard for evaluating QA systems.

**Weaknesses:**

The benchmark is tailored for KG-based QA tasks, which may not reflect the challenges present in more diverse, open-domain QA systems.
The reliance on GPT models for generating natural language representations from KGs may introduce subtle biases.
The rationale behind choosing specific complexity types (e.g., concatenation, intersection) could be expanded with examples illustrating real-world implications of these complexities in QA.

**Questions:**

1. How might CAQA be adapted to handle dynamic or temporal data in QA tasks?
2, What specific types of biases were identified or considered when using LLM-generated prompts?
3. Are there plans to include more diverse logical operations, such as negation, in future iterations of CAQA?
4. Could the inclusion of human-in-the-loop evaluations further enhance the quality of the generated benchmark data?
5, Extend the benchmark's applicability by including examples from various knowledge domains (e.g., medical, legal) to test the robustness of attributions in specialized contexts.
6. The reliance on GPT models for generating natural language representations from KGs may introduce subtle biases. Addressing how these biases are minimized or discussing potential implications would strengthen the manuscript.
7. Discussing how CAQA could be adapted for such tasks would add value especially how to address the challenges in more diverse, such as open-domain QA systems. .

---

> ### Author Response · Authors · 2024-12-01
> **Response - Part 1**
>
> Thank you for your detailed review and constructive feedback. We appreciate your insightful comments, which have provided valuable guidance for improving our work. Below, we address your main concerns by clarifying some misunderstandings.
> ## Weakness 1: The benchmark is tailored for KG-based QA tasks, which may not reflect the challenges present in more diverse, open-domain QA systems.
> **Response:** We believe there may be a misunderstanding regarding the scope and intent of our benchmark. Our work is not tailored specifically for KG-based QA tasks but rather aims to simulate the QA process of open-domain QA systems, such as Bing Chat. Specifically, our benchmark focuses on attributed QA, where the system is required to generate an answer to a given question while citing relevant evidence to support its answer.
>
> To construct this attributed QA benchmark in a scalable and automated manner, we propose methods that leverage knowledge graphs (KGs) and KG-based QA datasets as a foundation. These methods are designed to ensure that the benchmark reflects the challenges of generating accurate, evidence-supported answers in open-domain QA settings.
> ## Weakness 2: The reliance on GPT models for generating natural language representations from KGs may introduce subtle biases.
> **Response:** We ensure the quality of the natural language text generated by focusing on grammatical coherence and content accuracy. Building on prior work [1], which demonstrates that ChatGPT excels in KG-to-Text tasks with high grammatical correctness and coherence, we primarily evaluate content accuracy to ensure consistency with the corresponding triples.
> Specifically, we adopted the evaluation framework outlined in [1], assessing whether the generated text accurately reflects the triples. We randomly sampled 100 examples and employed two independent annotators to label each instance according to one of three exclusive categories:
>
> 1. **Full Coverage**: The text fully and correctly states all triples.
>
> 2. **Absent**: The text misses some triples.
>
> 3. **Hallucinated**: The text introduces content that actively contradicts the triples.
>
> The results are as follows:
>
> | Annotator  | Full Coverage |  Absent | Hallucinated |
> |------|------|------|------|
> | **A** | 92 | 5 | 3 |
> |**B** | 95 |3 |2 |
>
> The Cohen's Kappa score is 0.758, indicating substantial agreement between two annotators. The annotation results demonstrate that ChatGPT reliably generates accurate and coherent text within our benchmark.
>
> ## Weakness 3: The rationale behind choosing specific complexity types (e.g., concatenation, intersection) could be expanded with examples illustrating real-world implications of these complexities in QA.
> **Response:** The inclusion of specific complexity types, such as concatenation and intersection, in our benchmark is driven by their prevalence and practical significance in real-world open-domain QA scenarios. These complexity types arise frequently when answering questions that require reasoning across multiple documents or combining information from disparate sources to derive a correct answer.
>
> To illustrate this point, we provide examples from the widely used HotpotQA dataset [2], which contains such complexities:
>
> | Reasoning Type  | Example(s) |
> |------|------|
> | Type I (chain reasoning, a.k.a concatenation) | **Document A**: The *2015 Diamond Head Classic* was a college basketball tournament ... *Buddy Hield was named the tournament’s MVP*. <br>**Document B**: *Chavano Rainier ”Buddy” Hield* is a Bahamian professional basketball player for the Sacramento Kings of the NBA... <br> **Question**: Which team does the player named *2015 Diamond Head Classic’s MVP* play for? |
> |Type II (multiple properties, a.k.a intersection) | **Document A**: Several current and former members of *the Pittsburgh Pirates* ... John Milner, **Dave Parker**, and Rod Scurry... <br> **Document B**: **David Gene Parker**, *nicknamed ”The Cobra”*, is an American former player in Major League Baseball... <br> **Question**: Which former member of *the Pittsburgh Pirates* was *nicknamed ”The Cobra”*? |
>
> [1] Axelsson, Agnes, and Gabriel Skantze. "Using Large Language Models for Zero-Shot Natural Language Generation from Knowledge Graphs." Proceedings of the Workshop on Multimodal, Multilingual Natural Language Generation and Multilingual WebNLG Challenge (MM-NLG 2023). 2023.
>
> [2] Yang Z, Qi P, Zhang S, et al. HotpotQA: A Dataset for Diverse, Explainable Multi-hop Question Answering. Proceedings of the 2018 Conference on Empirical Methods in Natural Language Processing. Association for Computational Linguistics, 2018.

---

> ### Author Response · Authors · 2024-12-01
> **Response - Part 2**
>
> ## Question 1: How might CAQA be adapted to handle dynamic or temporal data in QA tasks?
>
> **Response:**  It appears there may be a misunderstanding regarding the scope of the CAQA benchmark. The CAQA benchmark is designed specifically to simulate the QA process of the attributed QA systems like Bing Chat, focusing on evaluating and developing more effective attribution evaluators. It is not inherently aimed at addressing dynamic or temporal data in QA tasks.
>
> Nonetheless, we believe the CAQA construction method can be extended or adapted to dynamic or temporal data. However, such adaptations would likely require temporal or evolving KGs.
>
> To clarify the current application of CAQA, we direct your attention to Section 6.4 of the paper, where we demonstrate that attribution evaluators developed using the CAQA benchmark outperform existing evaluators when applied to the actual attribution QA benchmark. This supports the validity and effectiveness of CAQA in its intended domain.
>
> ## Question 2: What specific types of biases were identified or considered when using LLM-generated prompts?
>
> **Response:** When designing LLM-generated prompts, we prioritized coherence and factual accuracy, ensuring that the generated content fully reflects the information in the corresponding triples. The designed prompt is presented in Table 9 of Appendix A.
> Additionally, we utilized the evaluation framework outlined in [1] to assess whether the generated text accurately represents the triples. The results show that our designed prompts enable the LLM to reliably produce accurate text within our benchmark. Details of the evaluation process are shown in Response Weakness 2.
>
> ## Question 3: Are there plans to include more diverse logical operations, such as negation, in future iterations of CAQA?
>
> **Response:** We acknowledge the importance of supporting a broader range of logical operations, including negation, to enhance the diversity and complexity of CAQA. In future iterations, we plan to incorporate additional logical operations such as Negation and Kleene Plus (+) in the Query Extension stage (See Line 323). In the current version, we focused on a subset of logical operations that are more prevalent in real-world QA scenarios.
>
> ## Question 4: Could the inclusion of human-in-the-loop evaluations further enhance the quality of the generated benchmark data?
>
> **Response:** To enhance quality of the generated benchmark data, we have incorporated human-in-the-loop evaluations during the prompt design phase for LLM-generated data. Specifically, we employ multiple prompts to guide the LLM in generating text, and then the generated output is evaluated by two annotators. These annotators assessed the quality of the text based on its alignment with the corresponding triples, utilizing the evaluation framework described in [1]. This process enabled us to refine and select higher-quality prompts, ultimately improving the generated benchmark data. Details of the evaluation process are elaborated in our response to Weakness 2.
>
> In future iterations of CAQA, we plan to expand the involvement of human evaluators, incorporating additional steps to further enhance the quality and reliability of the benchmark data.

---

> ### Author Response · Authors · 2024-12-01
> **Response - Part 3**
>
> ## Question 5: Extend the benchmark's applicability by including examples from various knowledge domains (e.g., medical, legal) to test the robustness of attributions in specialized contexts.
> **Response:**  In fact, we have constructed a domain attributed QA benchmark based on the domain KG (e.g., DBLP academic KG [3]) and have evaluated various attribution evaluators on this benchmark. While this contribution was omitted from the paper due to space constraints, we have made the benchmark and relevant evaluation results publicly available on the GitHub repository linked in the paper. We believe this addition complements the work and enhances its utility across specialized domains.
>
> The statistics for the new benchmark are as follows:
>
> | Classes  |  |  Train | Test | Total |
> |------|------|------|------|------|
> | | | 6,240 |1,748 |7,988|
> |Category |Sup. |1,735 |487 |2,222 |
> | | Ins. |1035 |287 |1,322 |
> | |Con. |1,735 |487 |2,222|
> | |Irr. |1,735 |487 |2,222|
> | Complexity| S.| 2,100| 600 |2,700 |
> | | C.| 500 |172 |672 |
> | |U. | 1,340 |348 |1,688|
> | |I. | 2,300| 628 |2,928|
>
> The experiment results of various evaluators on the new domain Attributed QA benchmarks are as follow:
>
> | Settings  | Evaluators(Size)  | | | | | Category  |   |   |  | Complexity |
> |------|------|------|------|------|------|------|------|------|------|------|
> |      |      |   Sup.  | Ins.  | Con.  | Irr.  | Overall  | S.  | C.  | I.  | U.  |
> | Zero-Shot  | Llama-2 (7B)  | 0.426 |0.130 | 0.026 |0.103 |0.274 |0.303 |0.276 |0.262 |0.247 |
> | |llama-3 (8B) |0.453 |0.100 |0.040 |0.000 |0.289 |0.328 |0.253 |0.258 |0.299 |
> | |Vicuna (7B)| 0.465 | 0.125 |0.090|0.030|0.305|0.360 |0.281 |0.268 |0.291 |
> | |GPT-3.5-turbo| 0.517 |0.135 |0.231 |0.129 |0.350 |0.420 |0.326 |0.291 |0.348|
> | | GPT-4 |**0.644**|**0.444** |**0.487** |**0.490** |**0.540** |**0.56** |**0.512** |**0.451** |**0.681** |
> | Few-Shot | Llama-2 (7B)| 0.353 |0.020 |0.058 |0.277 |0.252 |0.289 |0.295 |0.239 |0.188 |
> | | Llama-3 (8B) | 0.347 | 0.020 | 0.057 | 0.278 | 0.249 | 0.279 | 0.276| 0.253| 0.192|
> | | Vicuna (7B)| 0.325 |0.083 |0.000 |0.329 |0.256 | 0.338 |0.071 |0.250| 0.167|
> | |GPT-3.5-turbo | 0.541 |0.310| 0.112 |0.356 |0.401 |0.433 |0.390 |0.341 |0.460
> | |GPT-4 |**0.657** |**0.502** |**0.504** |**0.531** |**0.567** |**0.590** |**0.500** |**0.489** |**0.701**|
> | Fine-Tuning| Llama-2 (7B)| **0.733** |0.799 |0.712 |0.982 |0.801 |0.922 |**0.739** |**0.672** |0.888 |
> | | Llama-3 (8B) |0.726 |**0.847** |**0.715** |0.974 |**0.813** |**0.937** |0.692 |0.659 |**0.937** |
> | | Vicuna (7B) |0.693 |0.771 |0.675 |**0.989**  |0.783 |0.917 |0.709 |0.623 |0.876|
> | |AutoIS (11B) |0.410 |- |- |- |- |- |- |- |-|
> | | AttrScore (13B) | 0.545 |- |0.374 |0.488 |0.442 |0.553 |0.343 |0.376 |0.417|
>
> The conclusions are generally consistent with that on the CAQA benchmark.
> ## Question 6: The reliance on GPT models for generating natural language representations from KGs may introduce subtle biases. Addressing how these biases are minimized or discussing potential implications would strengthen the manuscript.
> **Response:** To minimize subtle biases and ensure the quality of generated text with GPT models, we employed the evaluation framework described in [1] and employed two annotators to assess text quality. This process allowed us to refine and select higher-quality prompts for the models. The annotation results confirm that GPT models reliably produce accurate and coherent text within our benchmark. Further details on the evaluation process are provided in our response to Weakness 2.
> ## Question 7: Discussing how CAQA could be adapted for such tasks would add value especially how to address the challenges in more diverse, such as open-domain QA systems.
> **Response:** The attribution evaluator developed by our benchmark can indeed be adapted for open-domain QA systems, particularly within the Retrieval-Augmented Generation (RAG) paradigm. Our benchmark evaluates existing attribution methods and provides a foundation for developing more refined evaluators. These refined evaluators can offer continuous performance measurement for QA systems, such as Bing Chat, while delivering fine-grained feedback to enhance various components of RAG systems. For instance, they can guide processes like re-retrieval, re-generation, and filtering irrelevant content from retrieved data. These enhancements address key challenges such as improving factuality, faithfulness, and reducing hallucination issues (Lines 45–48).
>
> While we briefly discussed these applications in the current version of the paper due to space constraints, we acknowledge the value of elaborating further. In a future version, we will include a more comprehensive discussion on adapting our contributions to open-domain QA systems and addressing their challenges.
>
> [3] Debayan B., Sushil A., et al. 2023. DBLP-QuAD: A Question Answering Dataset over the DBLP Scholarly Knowledge Graph. In Proceedings of the 13th International Workshop on Bibliometric-enhanced.

---

> ### Author Response · Authors · 2024-12-03
>
> We sincerely appreciate the valuable feedback and time you have dedicated to reviewing our work. We hope that our responses have addressed your concerns and provided clarity on the points raised.
>
> If there are any additional questions, suggestions, or clarifications needed, we would be happy to provide further information to ensure all concerns are fully addressed.

---

### Official Review · Reviewer_7BAF · 2024-11-03

**Soundness:** 3
**Presentation:** 3
**Contribution:** 3
**Rating:** 5
**Confidence:** 4

**Summary:**

The paper introduces the dataset Complex Attributed Question Answering (CAQA), containing answers to questions with associated source attributions, where the attributions may or may not support the answer. The non-support attributions are divided into 3 labeled categories: Partially supported, Contradictory Irrelevant.

They evaluate how well different LLMs can classify Q+A+source into these 4 categories, finding that in many cases they struggle to do well, especially on distinguishing the non-supportive categories.

The CAQA dataset is constructed from existing KGQA datasets (GrailQA and WebQuestionsSP), making use of the associated knowledge graph to produce different types of non-supportive evidence (and using GPT-3.5 to turn KG triples into natural language sentences). The resulting dataset is quite big (137k train, 24k test), allowing for fine-tuning experiments as well. The fine-tuned models do very well in distribution, and they also do limited out-of-distribution evaluation on a subset of ALCE further annotated with these non-supportive categories, showing promising results there as well.

**Strengths:**

The dataset is relevant for the important topic of answers with attributions from LLMs. Being able to carefully validate whether an answer actually follows from the sources is an important skill, and this dataset aims at helping with this.

The paper is well written, clearly describing the approach.

The use of the KG to create various incorrect attributions, together with using LLM to rewrite at text, seems quite effective.

The paper provides access to the full dataset for exploration which is truly helpful in assessing it.

The methods are tested on a more realistic, OOD, dataset.

**Weaknesses:**

While breaking down the non-supportive cases into three subcategories can be helpful for understanding limitations, the boundary between them can be quite unclear. Also the prompt for the non-GPT models doesn't go into great detail (beyond some examples) on what each category means. For instance, the "contradictory" evidence is often for actual true facts, so they're not actually contradiction, it's just the "wrong" evidence.

E.g., the answer "The person who founded the United States Coast Guard also founded the United States Department of the Treasury." is presented as being contradicted by the source "Alexander Hamilton is the founder of the United States Coast Guard and the Montgomery County Sheriff's Office, which is a government agency.", but this isn't really a contradiction, it's more like missing evidence. A true contradiction should lead you to think the answer is actually false, if you trust the source.

The "Partial support" category also can be quite subjective, as in the case of "The 2011 Estoril Open tournament event competition is the Men's Singles." being partially supported by "The 2011 Estoril Open had a men's singles competition."  (what's missing is apparently that "2011 Estoril Open" was "tournament event competition", but that's pretty much implied by the fact that they had a men's single compeition).

Because of this, it might also be useful to report the most important "supported" vs "not supported" scores.

Another concern is the simplicity of the dataset, with simple QA assertions attributed by short source sentences. How does good performance on CAQA transfer to more realistic settings. And can it be used to train better source attribution models as well? There is some exploration of this with the OOD ALCE dataset, but the effect (e.g., between Vicuna-13B and Vicuna-13B-finetuned) isn't as impactful as one might have hoped.

**Questions:**

Some discussion on the ambiguity (and actual errors) in the labeled categories would be useful (e.g., human annotator agreements on a sample).

Also would be good to discuss the lack of context consideration which is usually very important in real usage (e.g., the example in the paper "Who plays Fruma Sarah in Fiddler on the Roof" depends on which version of Fiddler on the Roof is being referenced).

---

> ### Author Response · Authors · 2024-12-01
> **Response - Part 1**
>
> Thank you for your detailed review and thoughtful feedback. We appreciate your recognition of our work's strengths and your constructive suggestions for improvement. In response to your concern that "the boundary between categories can be quite unclear," we have clarified the distinctions in the following response. We hope the response addresses your concerns about ambiguity and highlights the significance of these categories in understanding the system's limitations.
>
> ## Weakness 1: The "contradictory" evidence is often for actual true facts, so they're not actually contradiction, it's just the "wrong" evidence. The example isn't really a contradiction, it's more like missing evidence.
> **Response:** Regarding the reviewer's confusion about the difference between ‘*contradictory*’ and ‘*partially supported*’ (i.e., *missing*) cases, we need to clarify that the key difference lies in **whether the evidence can lead to an alternative answer to the question that directly conflicts with the generated answer.**
>
> * A case is labeled *contradictory* if the evidence leads to a new answer to the question, which is inconsistent with the generated answer.
>
> * A case is labeled *partially supported* if the evidence does not refute the generated answer but fails to provide sufficient support for it.
>
> To further clarify, we address the reviewer's example:
>
> **Question:** "The person who founded the United States Coast Guard also founded what government agency?"
>
> **Generated Answer:** "The person who founded the United States Coast Guard also founded the United States Department of the Treasury."
>
> **Evidence:** "Alexander Hamilton is the founder of the United States Coast Guard and the Montgomery County Sheriff's Office, which is a government agency."
>
> Here, the evidence clearly supports a new answer of "Montgomery County Sheriff's Office," which contradicts the generated answer. Thus, this case is categorized as "*contradictory*," not "*partially supported*," because the evidence can lead to a new answer to this question, which is clearly different from the generated answer.
>
> ## Weakness 2: The "Partial support" category also can be quite subjective, as in the case of "The 2011 Estoril Open tournament event competition is the Men's Singles." being partially supported by "The 2011 Estoril Open had a men's singles competition." (what's missing is apparently that "2011 Estoril Open" was "tournament event competition", but that's pretty much implied by the fact that they had a men's single competition).
> **Response:** We need to clarify the boundaries of "*Support*" and "*Partial support*". In our definition, "*Support*" requires evidence that explicitly and fully supports the generated answer without relying on implicit background knowledge or assumptions. In the specific example mentioned by the reviewer, while "The 2011 Estoril Open had a men's singles competition" provides contextual clues, it does not explicitly state that "The 2011 Estoril Open" was "tournament event competition". The classification as "*Partial support*" arises because the term "tournament event competition" is not explicitly tied to "men's singles" in the evidence. Inferring this connection would require background knowledge or assumptions about the nature of such events, which falls outside the scope of explicit *support*. This distinction helps maintain objectivity in our evaluation framework.
>
> ## Weakness 3: The prompt for the non-GPT models doesn't go into great detail (beyond some examples) on what each category means.
> **Response:** During our experiments, we have explored providing more detailed explanations for each category to non-GPT models. However, we observed that overly detailed prompts often led to a decline in their performance. This is likely due to the smaller parameter sizes of these models, which made them more susceptible to being overwhelmed or confused by excessive details. To maximize the performance of all models and to select the best attribution evaluator, we opted for prompts that were calibrated to achieve the best results for all models.

---

> ### Author Response · Authors · 2024-12-01
> **Response - Part 2**
>
> ## Weakness 4: Another concern is the simplicity of the dataset, with simple QA assertions attributed by short source sentences. How does good performance on CAQA transfer to more realistic settings. And can it be used to train better source attribution models as well? There is some exploration of this with the OOD ALCE dataset, but the effect (e.g., between Vicuna-13B and Vicuna-13B-finetuned) isn't as impactful as one might have hoped.
> **Response:** We appreciate the reviewer’s observation regarding the simplicity of the dataset and its implications for transferability to more realistic settings. While our benchmark primarily involves short evidence texts, it is designed to prioritize precision and granularity. Our approach enables the development of attribution evaluators that consistently outperform existing methods, as evidenced by the results in table 6 and table 8, thereby demonstrating the value of the proposed benchmark construction methodology.
>
> Regarding transferability, Table 8 demonstrates the performance of our attribution evaluators on a more challenging OOD dataset with longer evidence and answers. Notably, models developed using our benchmark achieve significant improvements: _T5-11B*_, trained with our dataset, achieves a 9% higher overall score compared to *AutoIS (T5-11B-based)*, and _Vicuna-13B*_, trained using our benchmark, achieves a 2% improvement over _AttrScore (Vicuna-13B-based)_. Additionally, our evaluators uniquely categorize attribution into four distinct types, offering deeper insights into model behavior compared to existing approaches.
>
> We also want to emphasize that our method for constructing attributed QA datasets is inherently scalable. By introducing noise into the attribution subgraph (Section 4.3), we can automatically generate longer and more complex evidence texts. This capability enhances the benchmark’s applicability to real-world scenarios.
>
> ## Question 1: Some discussion on the ambiguity (and actual errors) in the labeled categories would be useful (e.g., human annotator agreements on a sample).
> **Response:** Thank you for this valuable suggestion. We agree that discussing annotation ambiguities and errors is crucial for improving benchmark design and the robustness of attribution evaluators. To address this, we analyzed the labeling process and identified patterns of agreement and disagreement among annotators.
>
> Overall, the category with the highest agreement was ‘support’, followed by ‘irrelevant’, while the categories ‘partially supported’ and ‘contradictory’ exhibited lower agreement. Specifically:
>
> * Annotators often confused ‘partially supported’ with ‘irrelevant’, frequently misclassifying examples between these two categories. This confusion arose because distinguishing between them required identifying sub-facts (i.e., the smallest semantic units containing a subject, verb, and object) from answer statements and evidence. The process often required domain-specific knowledge, which annotators lacked, leading them to rely on co-occurring keywords rather than deeper semantic understanding.
>
> * Similarly, annotators had difficulty identifying ‘contradictory’ due to a lack of domain knowledge. This led to misclassifications of ‘contradictory’ as either ‘partially supported’ or ‘irrelevant’. Determining contradictions requires nuanced reasoning beyond surface-level overlaps, which posed challenges in the absence of sufficient domain context.
>
> ## Question 2: It would be good to discuss the lack of context consideration which is usually very important in real usage (e.g., the example in the paper "Who plays Fruma Sarah in Fiddler on the Roof" depends on which version of Fiddler on the Roof is being referenced).
> **Response:** This paper focuses solely on evaluating whether the generated answer is fully supported by the provided evidence, without tackling the broader challenge of contextual disambiguation. For example, if a QA system assumes *Fiddler on the Roof* refers to *the 1971 film* and generates the answer, "*xxx plays Fruma Sarah in the 1971 film Fiddler on the Roof*," the supporting evidence must explicitly reference "*the 1971 film Fiddler on the Roof*" to fully validate the answer.

---

> ### Author Response · Authors · 2024-12-03
>
> We sincerely appreciate the valuable feedback and time you have dedicated to reviewing our work. We hope that our responses have addressed your concerns and provided clarity on the points raised.
>
> If there are any additional questions, suggestions, or clarifications needed, we would be happy to provide further information to ensure all concerns are fully addressed.

---

> ### Author Response · Authors · 2024-12-04
>
> Thank you for your thoughtful feedback and for giving us the opportunity to address your concerns. We have made detailed clarifications and improvements based on your comments, including:
>
> * Clarifying definitions and boundaries for categories like "contradictory" and "partial support."
>
> * Clarifying the design motivation for the prompts of the non-GPT model.
>
> * Emphasizing the generalization advantages of our benchmark in realistic settings.
>
> * Expanding discussions on human annotations.
>
> We sincerely hope our responses address the concerns you raised. If you find the response satisfactory, we kindly ask you to consider revisiting the score you provided.
>
> We deeply appreciate your time and thoughtful insights, which have been instrumental in improving our work.

---

### Official Review · Reviewer_Eocw · 2024-11-04

**Soundness:** 3
**Presentation:** 4
**Contribution:** 3
**Rating:** 6
**Confidence:** 4

**Summary:**

This paper introduces Complex Attributed Question Answering (CAQA), a large-scale benchmark designed to evaluate complex attributions in question answering (QA). CAQA is automatically generated using knowledge graphs (KGs), includes a broader range of attribution categories along with intricate attribution reasoning scenarios, and is also aligned with human annotations. Experiments with two specifically developed evaluators and nine large language model (LLM) evaluators reveal that these models struggle to identify negative attribution categories and handle complex attribution reasoning in both zero-shot and few-shot settings but mostly perform relatively well in the fine-tuning setting.

**Strengths:**

1. This paper introduces CAQA, a large-scale benchmark for evaluating complex attributions in QA.
2. The CAQA dataset contains various new definitions (e.g., fine-grained attribute categories and attribution complexities), and the data construction process is automatic, considerate, and comprehensive.
3. This paper contains comprehensive experiments. In addition to model performance on CAQA, it also includes fine-grained analysis, human consistency, and out-of-distribution data.

**Weaknesses:**

1. This paper only considers GPT-3.5 and GPT-4 as closed-source LLMs, and some open-source LLMs used may be outdated (e.g., Mistral-7B has revolutionized various versions). Adding more diverse and latest models in experiments would have greater contributions and help to discover which LLMs perform best on this challenging task.
2. There is a lack of comparisons with human performance on (a subset) of the dataset, which would better illustrate the performance gap and the challenge of the dataset.
3. While the contribution of the paper centers on a new challenging benchmark, it would be much helpful if the authors can provide an error analysis, which will direct newcomers in future research.

**Questions:**

See "Weaknesses".

---

> ### Author Response · Authors · 2024-12-02
> **Response - Part 1**
>
> Thank you for your detailed review and constructive feedback on our submission. We appreciate your recognition of the strengths of our work and your thoughtful comments on areas where we could improve. Below, we address your main concerns:
> ## Weakness 1: Adding more diverse and latest models in experiments would have greater contributions and help to discover which LLMs perform best on this challenging task.
> **Response:** Thank you for pointing out the importance of including a broader range of models. To address this, we conducted additional experiments incorporating the latest LLMs. The results are as follows:
>
> | Settings  | Evaluators(Size)  |   |   |   |  | Category  |   |   |  | Complexity |
> |------|------|------|------|------|------|------|------|------|------|------|
> |      |      |   Sup.  | Ins.  | Con.  | Irr.  | Overall  | S.  | C.  | I.  | U.  |
> | **Zero-Shot**|Llama-3.1 (8B)| 0.54 | 0.05 | 0.13|0.02  |0.32 | 0.32|0.33 |0.32 | 0.28|
> | |Mistral-v0.3(7B)| 0.66 | 0.16  |0.05 |0.33  |0.36 |0.36 |0.37| 0.36 | 0.34|
> | |Phi-3-medium (14B)| 0.63 | 0.15  |0.38 | 0.29 | 0.40|0.41 |0.41 |0.41 |0.39 |
> | |Mixtral-v1.0 (8x7B)| 0.68 | 0.09 | 0.17 |0.63  | 0.49 |0.50 | 0.52| 0.48 |0.49 |
> | |Llama-3.1 (70B)| 0.69 | 0.17  | 0.55 | 0.61  | 0.54 | 0.54 | 0.55 |0.54| 0.50|
> | |Qwen2.5 (72B)| 0.63 | 0.27  | 0.70 |0.47  |0.57 | 0.57| 0.58 |0.56 |0.53 |
> | |GPT-4o| **0.77** | **0.44** | 0.60 |0.63 |**0.63** |**0.68** |**0.59** |**0.47** |**0.59**|
> | | GPT-4o-mini |0.72| 0.30 | **0.63** |**0.70** |0.62 |0.67 |0.47 |0.44 |0.56 |
> | **Few-Shot** |Llama-3.1 (8B)| 0.63 |0.10  |0.31 |0.16  |0.35 | 0.35 | 0.37|0.34 | 0.36|
> | |Mistral-v0.3(7B)|0.57  |0.08  |0.25 | 0.13 |0.34 | 0.34 | 0.35| 0.34|0.33 |
> | |Phi-3-medium (14B)| 0.63 | 0.12  |0.09 |0.45  |0.44 | 0.42| 0.44 |0.41 | 0.39|
> | |Mixtral-v1.0 (8x7B)| 0.59 |0.08  |0.36 |0.56  |0.46 | 0.46|0.47 |0.45 | 0.42|
> | |Llama-3.1 (70B)| 0.70 | 0.26  |0.71 | 0.66|0.63 | 0.63| **0.64**|**0.63** | **0.59**|
> | |Qwen2.5 (72B)| 0.72 | 0.40  | 0.74 |0.50  |0.62 |0.62 | 0.61| **0.63**| **0.59**|
> | |GPT-4o| **0.78** |**0.51** |0.68|0.64 |**0.66** |0.73 |0.56|0.45 |0.53|
> | | GPT-4o-mini |0.76|0.43 | **0.71**|**0.70** |**0.66** |**0.74** |0.43 |0.40 |**0.59**|
> | **Fine-Tuning**|Llama-3.1 (8B)| 0.94 | 0.92  | 0.94 |0.93  |0.94 |0.95 |0.85| 0.94 |0.94 |
> | |Mistral-v0.3(7B)| 0.94 |0.93  |0.94 |0.93| 0.94 |0.96 | 0.86 | 0.93 |0.94 |
>
> Based on the new experimental results, we have further made some new discoveries:
>
> 1. The newest LLMs demonstrate slight improvements in overall performance for attribution recognition compared to their earlier versions in all three settings. Specifically, in zero-shot and few-shot settings, they show more balanced and enhanced results across scenarios with varying levels of complexity, highlighting advancements in their reasoning capabilities. However, they still struggle to accurately distinguish fine-grained attribute categories, with notable deficiencies persisting in certain attribution recognition.
>
> 2. LLMs with larger parameter sizes (70B or above), such as Llama-3.1 (70B), Qwen-2.5 (72B), and GPT series models, exhibit significantly enhanced in-context learning capabilities. In few-shot settings, these models make significant improvements in distinguishing subtle differences between attribution categories, indicating that they have a deeper understanding of context.
>
> ## Weakness 2: There is a lack of comparisons with human performance on (a subset) of the dataset, which would better illustrate the performance gap and the challenge of the dataset.
> **Response:** We would like to clarify that the primary goal of our benchmark is to advance the development and evaluation of automated attribution evaluators, rather than to directly assess human performance as an attribution evaluator. Having humans act as attribution evaluators is both costly and inefficient, as described in lines 129–135. Instead, we follow previous work [1][2] and compare the consistency of human and benchmark annotations. Our experiments in Section 6.2 confirm that our benchmark is highly consistent with human assessments, which demonstrates its reliability for the intended purpose.
>
> [1] Gao T, Yen H, Yu J, et al. Enabling Large Language Models to Generate Text with Citations. Proceedings of the 2023 Conference on Empirical Methods in Natural Language Processing. 2023: 6465-6488.
>
> [2] Bohnet B, Tran V Q, Verga P, et al. Attributed question answering: Evaluation and modeling for attributed large language models[J]. arXiv preprint arXiv:2212.08037, 2022.

---

> ### Author Response · Authors · 2024-12-02
> **Response - Part 2**
>
> ## Weakness 3: While the contribution of the paper centers on a new challenging benchmark, it would be much helpful if the authors can provide an error analysis, which will direct newcomers in future research.
> **Response:** Thank you for your thoughtful suggestion. Due to space constraints, we have included key error analyses in Section 6.1 (lines 445–458), which highlight significant challenges faced even by the best-performing evaluators. We now provide a more detailed analysis of the representative evaluator, GPT-4o, focusing on areas where it struggles. Below, we summarize these findings and their implications:
>
> **1. Difficulty with the 'Partially Supported' Category:**
>
> GPT-4o achieves a recall of only 0.40 for this category, frequently misclassifying ‘partially supported’ cases as ‘supported.’ This issue arises from the model's reliance on keyword co-occurrence between the evidence and the answer, often favoring surface-level associations over deeper semantic understanding. This behavior reveals the model’s bias towards attribution evaluation, underscoring the need for models that can better capture nuanced semantic relationships.
>
> **2. Challenges with ‘Contradiction’ and ‘Irrelevant’ Categories:**
>
> GPT-4o exhibits a recall of 0.45 for the ‘contradiction’ category and a precision of 0.54 for the ‘irrelevant’ category. In complex scenarios (beyond "Single" complexity), the model struggles to discern subtle distinctions among erroneous attributions based solely on instructions. However, augmenting the model with a few demonstrations (few-shot setting) or fine-tuning significantly improves its performance in identifying these categories. This suggests that techniques such as in-context learning or fine-tuning may enhance the model’s attribution identification capabilities.

---

> ### Author Response · Authors · 2024-12-03
>
> We sincerely appreciate the valuable feedback and time you have dedicated to reviewing our work. We hope that our responses have addressed your concerns and provided clarity on the points raised.
>
> If there are any additional questions, suggestions, or clarifications needed, we would be happy to provide further information to ensure all concerns are fully addressed.

---

> ### Author Response · Authors · 2024-12-04
>
> Thank you for your thoughtful feedback on our submission. We greatly appreciate the opportunity to address your concerns and have made significant improvements based on your comments:
>
> * Expanding our experiments with a more diverse set of state-of-the-art LLMs and providing additional observations.
>
> * Clarifying the purpose of our benchmark, as well as detailing the use of human annotations.
>
> * Conducting a more detailed error analysis to offer insights that can guide future researchers.
>
> We sincerely hope that our revisions effectively address your concerns and enhance the quality of our work. If you find our responses satisfactory, we would greatly appreciate it if you could kindly reconsider the score you have provided.
>
> Thank you once again for your time and constructive feedback.

---

### Official Review · Reviewer_nZmL · 2024-11-04

**Soundness:** 3
**Presentation:** 3
**Contribution:** 3
**Rating:** 6
**Confidence:** 5

**Summary:**

The paper presents CAQA (Complex Attributed Question Answering), a large-scale automatically generated benchmark designed to assess the attribution capabilities of QA systems, particularly Large Language Models (LLMs). CAQA leverages Knowledge Graphs (KGs) to create comprehensive attribution categories and to handle complex reasoning scenarios. The benchmark distinguishes between supportive, partially supportive, contradictory, and irrelevant evidence types and introduces reasoning complexity through different forms of evidence combination (e.g., union, intersection, concatenation).

**Strengths:**

- CAQA uses KGs to generate complex QA benchmarks automatically, enabling scalability and minimizing manual annotation effort.
- Different reasoning complexities are considered, highlighting LLMs' capabilities in handling logical relationships between facts.
- The benchmark includes fine-grained attribution categories.

**Weaknesses:**

- The task setting seems very similar to NLI to me, more discussions are needed.
- Lack of a few details about the human annotation process.
- The distribution of the complexity is biased.

**Questions:**

- How do you verify the quality of converted natural language style questions?
- What is the inter-agreement score of human annotations?

---

> ### Author Response · Authors · 2024-12-01
> **Response - Part 1**
>
> We greatly appreciate the time and effort you have invested. In response to your concerns, we have provided clarifications here.
> ## Weakness 1: The task setting seems very similar to NLI to me, more discussions are needed.
> **Response:** We would like to clarify that the task of attribution evaluation in our paper introduces unique challenges that set it apart from standard NLI tasks.
>
> While NLI typically involves classifying relationships between two sentences into predefined categories such as entailment, contradiction, or neutral, our attribution evaluation benchmark defines four nuanced relationship categories between an answer and its cited evidence: support, partial support, contradiction, and irrelevant. This classification inherently requires finer-grained distinctions, particularly in the "partial support" category, which has no direct counterpart in traditional NLI.
> Moreover, our benchmark incorporates the complexity of reasoning across multiple pieces of cited evidence to determine their aggregated relationship to the answer. This multi-evidence reasoning poses a significant challenge to existing NLI models, as demonstrated by prior research [1], which highlights the limitations of AutoIS (an NLI model, *T5_xxl_true_nli_mixture* [2]) in scenarios involving multi-evidence reasoning (Lines 214-215).
>
> Our experimental results (Tables 6 and 8) further reinforce this distinction. AutoIS, despite being a state-of-the-art NLI model, struggles even with a simplified two-category (support/non-support) version of our task and performs poorly in practical attribution evaluation settings. These findings underscore that while our task draws from NLI principles, it extends beyond the traditional scope of NLI, addressing challenges that current NLI models are not capable of tackling.
> ## Weakness 2: Lack of a few details about the human annotation process.
> **Response:** We have clarified the details of the human annotation process in Appendix H. In response to the question regarding “‘What is the manually labelled in-protocol score?”, we provide the following additional details:
>
> For the CAQA benchmark, we sampled 400 cases and provided them to three annotators. Among these, 2 examples were deemed unclear by more than one annotator and subsequently discarded. This resulted in 398 valid annotated examples. To ensure reliability, we retained only examples where there was consistent agreement among the annotators (See Line 1063). The Fleiss' Kappa score for these annotations was 0.780, reflecting a substantial level of agreement.
> For the ALCE-FineGrained benchmark, we sampled 300 cases and provided them to three annotators. Of these, 24 examples were deemed unclear and were excluded, leaving 276 valid annotated examples. Similar to the CAQA benchmark, we retained only those examples with consistent annotations. The Fleiss' Kappa score for these annotations was 0.748, also indicating a substantial level of agreement.
> ## Weakness 3: The distribution of the complexity is biased.
> **Response:** We would like to clarify that our method is designed to allow flexible construction of sample sets across varying complexity levels, ensuring that a balanced distribution can be achieved when needed. However, our choice to employ an unbalanced complexity distribution in this work was intentional and aligns with the characteristics of real-world Attributed QA systems [3].
> In practice, most existing Attributed QA systems exhibit limited attribution capabilities, frequently citing a single piece of textual evidence to support answers (i.e., Single complexity). In contrast, only a small proportion of answers in these systems require synthesizing multiple independent facts and thus necessitate citing multiple pieces of evidence (i.e., Union complexity).
>
> To reflect this prevalent behavior, we designed our benchmark to mirror the complexity distribution typically observed in the outputs of Attributed QA systems. While this results in an unbalanced complexity distribution, it ensures the benchmark remains realistic and representative of current system performance. We believe this choice enhances the benchmark's relevance and applicability for evaluating such systems.
>
> [1] Malaviya C, Lee S, Chen S, et al. ExpertQA: Expert-Curated Questions and Attributed Answers. Proceedings of the 2024 Conference of the North American Chapter of the Association for Computational Linguistics: Human Language Technologies (Volume 1: Long Papers). 2024: 3025-3045.
>
> [2] Honovich O, Aharoni R, Herzig J, et al. TRUE: Re-evaluating Factual Consistency Evaluation. Proceedings of the 2022 Conference of the North American Chapter of the Association for Computational Linguistics: Human Language Technologies. 2022: 3905-3920.
>
> [3] Gao T, Yen H, Yu J, et al. Enabling Large Language Models to Generate Text with Citations. Proceedings of the 2023 Conference on Empirical Methods in Natural Language Processing. 2023: 6465-6488.

---

> ### Author Response · Authors · 2024-12-01
> **Response - Part 2**
>
> ## Question 1: How do you verify the quality of converted natural language style questions?
> **Response:** We ensure the quality of the natural language text generated by focusing on grammatical coherence and content accuracy. Building on prior work [4], which demonstrates that ChatGPT excels in KG-to-Text tasks with high grammatical correctness and coherence, we primarily evaluate content accuracy to ensure consistency with the corresponding triples.
>
> Specifically, we adopted the evaluation framework outlined in [4], assessing whether the generated text accurately reflects the triples. We randomly sampled 100 examples and employed two independent annotators to label each instance according to one of three exclusive categories:
>
> 1. **Full Coverage**: The text fully and correctly states all triples.
>
> 2. **Absent**: The text misses some triples.
>
> 3. **Hallucinated**: The text introduces content that actively contradicts the triples.
>
> The results are as follows:
> | Annotator  | Full Coverage |  Absent | Hallucinated |
> |------|------|------|------|
> | **A** | 92 | 5 | 3 |
> |**B** | 95 |3 |2 |
>
> The Cohen's Kappa score is 0.758, indicating substantial agreement between two annotators. The annotation results demonstrate that ChatGPT reliably generates accurate and coherent text within our benchmark.
>
> ## Question 2: What is the inter-agreement score of human annotations?
> **Response:** For the CAQA benchmark, the Fleiss' Kappa score for these annotations was 0.780, reflecting a substantial level of agreement. For the ALCE-FineGrained benchmark, the Fleiss' Kappa score for these annotations was 0.748, also indicating a substantial level of agreement. For a detailed explanation, see response to Weakness 2.
>
> [4] Axelsson, Agnes, and Gabriel Skantze. "Using Large Language Models for Zero-Shot Natural Language Generation from Knowledge Graphs." Proceedings of the Workshop on Multimodal, Multilingual Natural Language Generation and Multilingual WebNLG Challenge (MM-NLG 2023). 2023.

---

> ### Comment · Reviewer_nZmL · 2024-12-03
>
> Thank you for the response, I'm still ambiguous about the novelty & challenge of the task, especially when considering LLM becomes stronger. I increased my overall rating by 1.

---

> > ### Author Response · Authors · 2024-12-03
> >
> > We sincerely appreciate your feedback and the increased rating.
> > # Regarding novelty
> > We would like to highlight the distinct contributions of our benchmark compared to prior work. Unlike previous benchmarks (as outlined in Table 1), our benchmark offers the following advantages:
> >
> > * **Automatic construction**: Reduces manual effort and ensures scalability.
> >
> > * **Larger scale**: Significantly surpasses previous benchmarks in size.
> >
> > * **Broader coverage**: Includes more fine-grained categories.
> >
> > * **Complexity classification**: Introduces a novel dimension by categorizing tasks based on complexity, enabling a detailed analysis of model performance across varying difficulty levels.
> >
> > # Regarding the challenge of the task
> > We have conducted additional experiments with the latest state-of-the-art LLMs. Despite their improved capabilities, these models still struggle to perform well on our benchmark, indicating that our task remains challenging and highlights areas where even advanced models fall short. The results are as follows:
> >
> > | Settings  | Evaluators(Size)  |   |   |   |  | Category  |   |   |  | Complexity |
> > |------|------|------|------|------|------|------|------|------|------|------|
> > |      |      |   Sup.  | Ins.  | Con.  | Irr.  | Overall  | S.  | C.  | I.  | U.  |
> > | **Zero-Shot**|Llama-3.1 (8B)| 0.54 | 0.05 | 0.13|0.02  |0.32 | 0.32|0.33 |0.32 | 0.28|
> > | |Mistral-v0.3(7B)| 0.66 | 0.16  |0.05 |0.33  |0.36 |0.36 |0.37| 0.36 | 0.34|
> > | |Phi-3-medium (14B)| 0.63 | 0.15  |0.38 | 0.29 | 0.40|0.41 |0.41 |0.41 |0.39 |
> > | |Mixtral-v1.0 (8x7B)| 0.68 | 0.09 | 0.17 |0.63  | 0.49 |0.50 | 0.52| 0.48 |0.49 |
> > | |Llama-3.1 (70B)| 0.69 | 0.17  | 0.55 | 0.61  | 0.54 | 0.54 | 0.55 |0.54| 0.50|
> > | |Qwen2.5 (72B)| 0.63 | 0.27  | 0.70 |0.47  |0.57 | 0.57| 0.58 |0.56 |0.53 |
> > | |GPT-4o| **0.77** | **0.44** | 0.60 |0.63 |**0.63** |**0.68** |**0.59** |**0.47** |**0.59**|
> > | | GPT-4o-mini |0.72| 0.30 | **0.63** |**0.70** |0.62 |0.67 |0.47 |0.44 |0.56 |
> > | **Few-Shot** |Llama-3.1 (8B)| 0.63 |0.10  |0.31 |0.16  |0.35 | 0.35 | 0.37|0.34 | 0.36|
> > | |Mistral-v0.3(7B)|0.57  |0.08  |0.25 | 0.13 |0.34 | 0.34 | 0.35| 0.34|0.33 |
> > | |Phi-3-medium (14B)| 0.63 | 0.12  |0.09 |0.45  |0.44 | 0.42| 0.44 |0.41 | 0.39|
> > | |Mixtral-v1.0 (8x7B)| 0.59 |0.08  |0.36 |0.56  |0.46 | 0.46|0.47 |0.45 | 0.42|
> > | |Llama-3.1 (70B)| 0.70 | 0.26  |0.71 | 0.66|0.63 | 0.63| **0.64**|**0.63** | **0.59**|
> > | |Qwen2.5 (72B)| 0.72 | 0.40  | 0.74 |0.50  |0.62 |0.62 | 0.61| **0.63**| **0.59**|
> > | |GPT-4o| **0.78** |**0.51** |0.68|0.64 |**0.66** |0.73 |0.56|0.45 |0.53|
> > | | GPT-4o-mini |0.76|0.43 | **0.71**|**0.70** |**0.66** |**0.74** |0.43 |0.40 |**0.59**|
> > | **Fine-Tuning**|Llama-3.1 (8B)| 0.94 | 0.92  | 0.94 |0.93  |0.94 |0.95 |0.85| 0.94 |0.94 |
> > | |Mistral-v0.3(7B)| 0.94 |0.93  |0.94 |0.93| 0.94 |0.96 | 0.86 | 0.93 |0.94 |
> >
> > Based on the new experimental results, we have further made some new discoveries:
> >
> > 1. The newest LLMs demonstrate slight improvements in overall performance for attribution recognition compared to their earlier versions in all three settings. Specifically, in zero-shot and few-shot settings, they show more balanced and enhanced results across scenarios with varying levels of complexity, highlighting advancements in their reasoning capabilities. However, they still struggle to accurately distinguish fine-grained attribute categories, with notable deficiencies persisting in certain attribution recognition.
> >
> > 2. LLMs with larger parameter sizes (70B or above), such as Llama-3.1 (70B), Qwen-2.5 (72B), and GPT series models, exhibit significantly enhanced in-context learning capabilities. In few-shot settings, these models make significant improvements in distinguishing subtle differences between attribution categories, indicating that they have a deeper understanding of context.

---

### Meta-Review · Area_Chair_iMXa · 2024-12-23

**Metareview:**

This paper presents options for evaluating complex attributed question answering tasks.  I think this paper can't be accepted due to multiple reasons:

1.  I think the "partially supported" category does not make any sense for the attribution task, which was originally proposed by Rashkin et al (2023).  Either a claim in a response is attributable to evidence or not.  Hence, focusing too much on that category--which this paper does, weakens its experimental design.  Several reviewers point this out.

2.  It is unclear to me whether the paper will generalize to tasks beyond ones that rely on KGs--another point mentioned by reviewers.

It will be great to address these issues in addition to the finer grained issues mentioned below, to improve the draft.

**Additional Comments On Reviewer Discussion:**

There is significant back and forth between reviewers and authors for this paper.

---

### Decision · Program_Chairs · 2025-01-22

Reject